# DATA-DRIVEN LEARNING OF GEOMETRIC SCATTERING NETWORKS

## ABSTRACT

Many popular graph neural network (GNN) architectures, which are often considered as the current state of the art, rely on encoding graph structure via smoothness or similarity between neighbors. While this approach performs well on a surprising number of standard benchmarks, the efficacy of such models does not translate consistently to more complex domains, such as graph data in the biochemistry domain. We argue that these more complex domains require priors that encourage learning of longer range features rather than oversmoothed signals of standard GNN architectures. Here, we propose an alternative GNN architecture, based on a relaxation of recently proposed geometric scattering transforms, which consists of a cascade of graph wavelet filters. Our learned geometric scattering (LEGS) architecture adaptively tunes these wavelets and their scales to encourage band-pass features to emerge in learned representations. This results in a simplified GNN with significantly fewer learned parameters compared to competing methods. We demonstrate the predictive performance of our method on several biochemistry graph classification benchmarks, as well as the descriptive quality of its learned features in biochemical graph data exploration tasks. Our results show that the proposed LEGS network matches or outperforms popular GNNs, as well as the original geometric scattering construction, while retaining certain mathematical properties of its handcrafted (nonlearned) design.

## 1 INTRODUCTION

Geometric deep learning has recently emerged as an increasingly prominent branch of machine learning in general, and deep learning in particular (Bronstein et al., 2017). It is based on the observation that many of the impressive achievements of neural networks come in applications where the data has an intrinsic geometric structure which can be used to inform network design and training procedures. For example, in computer vision, convolutional neural networks use the spatial organization of pixels to define convolutional filters that hierarchically aggregate local information at multiple scales that in turn encode shape and texture information in data and task-driven representations. Similarly, in time-series analysis, recurrent neural networks leverage memory mechanisms based on the temporal organization of input data to collect multiresolution information from local subsequences, which can be interpreted geometrically via tools from dynamical systems and spectral analysis. While these examples only leverage Euclidean spatiotemporal structure in data, they exemplify the potential benefits of incorporating information about intrinsic data geometry in neural network design and processing. Indeed, recent advances have further generalized the utilization of geometric information in neural networks design to consider non-Euclidean structures, with particular interest in graphs that represent data geometry, either directly given as input or constructed as an approximation of a data manifold.

At the core of geometric deep learning is the use of graph neural networks (GNNs) in general, and graph convolutional networks (GCNs) in particular, which ensure neuron activations follow the geometric organization of input data by propagating information across graph neighborhoods (Bruna et al., 2014; Defferrard et al., 2016; Kipf & Welling, 2016; Hamilton et al., 2017; Xu et al., 2019; Abu-El-Haija et al., 2019). However, recent work has shown the difficulty in generalizing these methods to more complex structures, identifying common problems and phrasing them in terms of oversmoothing (Li et al., 2018), oversquashing (Alon & Yahav, 2020) or under-reaching (Barceló et al., 2020). Using graph signal processing terminology from Kipf & Welling (2016), these issues

can be partly attributed to the limited construction of convolutional filters in many commonly used GCN architectures. Inspired by the filters learned in convolutional neural networks, GCNs consider node features as graph signals and aim to aggregate information from neighboring nodes. For example, Kipf & Welling (2016) presented a typical implementation of a GCN with a cascade of averaging (essentially low pass) filters. We note that more general variations of GCN architectures exist (Defferrard et al., 2016; Hamilton et al., 2017; Xu et al., 2019), which are capable of representing other filters, but as investigated in Alon & Yahav (2020), they too often have difficulty in learning long range connections.

Recently, an alternative approach was presented to provide deep geometric representation learning by generalizing Mallat's scattering transform (Mallat, 2012), originally proposed to provide a mathematical framework for understanding convolutional neural networks, to graphs (Gao et al., 2019; Gama et al., 2019a; Zou & Lerman, 2019) and manifolds (Perlmutter et al., 2018). Similar to traditional scattering, which can be seen as a convolutional network with nonlearned wavelet filters, geometric scattering is defined as a GNN with handcrafted graph filters, typically constructed as diffusion wavelets over the input graph (Coifman & Maggioni, 2006), which are then cascaded with pointwise absolute-value nonlinearities. This wavelet cascade results in permutation equivariant node features that are typically aggregated via statistical moments over the graph nodes, as explained in detail in Sec. 2, to provide a permutation invariant graph-level representation. The efficacy of geometric scattering features in graph processing tasks was demonstrated in Gao et al. (2019), with both supervised learning and data exploration applications. Moreover, their handcrafted design enables rigorous study of their properties, such as stability to deformations and perturbations, and provides a clear understanding of the information extracted by them, which by design (e.g., the cascaded band-pass filters) goes beyond low frequencies to consider richer notions of regularity (Gama et al., 2019b; Perlmutter et al., 2019).

However, while graph scattering transforms provide effective universal feature extractors, their rigid handcrafted design does not allow for the automatic task-driven representation learning that naturally arises in traditional GNNs. To address this deficiency, recent work has proposed a hybrid scattering-GCN (Min et al., 2020) model for obtaining node-level representations, which ensembles a GCN model with a fixed scattering feature extractor. In Min et al. (2020), integrating channels from both architectures alleviates the well-known oversmoothing problem and outperforms popular GNNs on node classification tasks. Here, we focus on improving the geometric scattering transform by learning, in particular its scales. We focus on whole-graph representations with an emphasis on biochemical molecular graphs, where relatively large diameters and non-planar structures usually limit the effectiveness of traditional GNNs. Instead of the ensemble approach of Min et al. (2020), we propose a native neural network architecture for learned geometric scattering (LEGS), which directly modifies the scattering architecture from Gao et al. (2019); Perlmutter et al. (2019), via relaxations described in Sec. 3, to allow a task-driven adaptation of its wavelet configuration via backpropagation implemented in Sec. 4. We note that other recent graph spectrum-based methods approach the learning of long range connections by approximating the spectrum of the graph with the Lancoz algorithm Liao et al. (2019), or learning in block Krylov subspaces Luan et al. (2019). Such methods are complementary to the work presented here, in that their spectral approximation can also be applied in the computation of geometric scattering when considering very long range scales (e.g., via spectral formulation of graph wavelet filters). However, we find that such approximations are not necessary in the datasets considered here and in other recent work focusing on whole-graph tasks, where direct computation of polynomials of the Laplacian is sufficient.

The resulting learnable geometric scattering network balances the mathematical properties inherited from the scattering transform (as shown in Sec. 3) with the flexibility enabled by adaptive representation learning. The benefits of our construction over standard GNNs, as well as pure geometric scattering, are discussed and demonstrated on graph classification and regression tasks in Sec. 5. In particular, we find that our network maintains the robustness to small training sets present in graph scattering while improving classification on biological graph classification and regression tasks, and we show that in tasks where the graphs have a large diameter relative to their size, learnable scattering features improve performance over competing methods.

## 2  PRELIMINARIES: GEOMETRIC SCATTERING FEATURES

Let $\mathcal{G} = (V, E, w)$ be a weighted graph with $V := \{v_1, \ldots, v_n\}$ the set of nodes, $E \subset \{\{v_i, v_j\} \in V \times V, i \neq j\}$ the set of (undirected) edges and $w : E \to (0, \infty)$ assigning (positive) edge weights to the graph edges. Note that $w$ can equivalently be considered as a function of $V \times V$, where we set the weights of non-adjacent node pairs to zero. We define a *graph signal* as a function $x : V \to \mathbb{R}$ on the nodes of $\mathcal{G}$ and aggregate them in a signal vector $\boldsymbol{x} \in \mathbb{R}^n$ with the $i^{th}$ entry being $x[v_i]$.

We define the *weighted adjacency matrix* $\boldsymbol{W} \in \mathbb{R}^{n \times n}$ of the graph $\mathcal{G}$ as

$$W[v_i, v_j] := \begin{cases} w(v_i, v_j) & \text{if } \{v_i, v_j\} \in E \\ 0 & \text{otherwise} \end{cases},$$

and the *degree matrix* $\boldsymbol{D} \in \mathbb{R}^{n \times n}$ of $\mathcal{G}$ as $\boldsymbol{D} := \text{diag}(d_1, \ldots, d_n)$ with $d_i := \deg(v_i) := \sum_{j=1}^n W[v_i, v_j]$ being the *degree* of the node $v_i$.

The geometric scattering transform (Gao et al., 2019) relies on a cascade of graph filters constructed from a row stochastic diffusion matrix $\boldsymbol{P} := \frac{1}{2}(\boldsymbol{I}_n + \boldsymbol{W}\boldsymbol{D}^{-1})$, which corresponds to transition probabilities of a lazy random walk Markov process. The laziness of the process signifies that at each step it has equal probability of either staying at the current node or transitioning to a neighbor, where transition probabilities in the latter case are determined by (normalized) edge weights. Scattering filters are then defined via the graph-wavelet matrices $\boldsymbol{\Psi}_j \in \mathbb{R}^{n \times n}$ of scale $j \in \mathbb{N}_0$, as

$$\boldsymbol{\Psi}_0 := \boldsymbol{I}_n - \boldsymbol{P},$$
$$\boldsymbol{\Psi}_j := \boldsymbol{P}^{2^{j-1}} - \boldsymbol{P}^{2^j} = \boldsymbol{P}^{2^{j-1}}(\boldsymbol{I}_n - \boldsymbol{P}^{2^{j-1}}), \quad j \geq 1. \tag{1}$$

These diffusion wavelet operators partition the frequency spectrum into dyadic frequency bands, which are then organized into a full wavelet filter bank $\mathcal{W}_J := \{\boldsymbol{\Psi}_j, \boldsymbol{\Phi}_J\}_{0 \leq j \leq J}$, where $\boldsymbol{\Phi}_J := \boldsymbol{P}^{2^J}$ is a pure low-pass filter, similar to the one used in GCNs. It is easy to verify that the resulting wavelet transform is invertible, since a simple sum of filter matrices in $\mathcal{W}_J$ yields the identity. Moreover, as discussed in Perlmutter et al. (2019), this filter bank forms a nonexpansive frame, which provides energy preservation guarantees as well as stability to perturbations, and can be generalized to a wider family of constructions that encompasses the variations of scattering transforms on graphs from Gama et al. (2019a;b) and Zou & Lerman (2019).

Given the wavelet filter bank $\mathcal{W}_J$, node-level scattering features are computed by stacking cascades of bandpass filters and element-wise absolute value nonlinearities to form

$$\boldsymbol{U}_p\boldsymbol{x} := \boldsymbol{\Psi}_{j_m}|\boldsymbol{\Psi}_{j_{m-1}} \ldots |\boldsymbol{\Psi}_{j_2}|\boldsymbol{\Psi}_{j_1}\boldsymbol{x}|| \ldots |, \tag{2}$$

indexed (or parametrized) by the scattering path $p := (j_1, \ldots, j_m) \in \cup_{m \in \mathbb{N}} \mathbb{N}_0^m$ that determines the filter scales captured by each scattering coefficient. Then, a whole-graph scattering representation is obtained by aggregating together node-level features via statistical moments over the nodes of the graph (Gao et al., 2019). This construction yields the geometric scattering features

$$\boldsymbol{S}_{p,q}\boldsymbol{x} := \sum_{i=1}^n |\boldsymbol{U}_p\boldsymbol{x}[v_i]|^q. \tag{3}$$

indexed by the scattering path $p$ and moment order $q$. Finally, we note that it can be shown that the graph-level scattering transform $\boldsymbol{S}_{p,q}$ guarantees node-permutation invariance, while $\boldsymbol{U}_p$ is permutation equivariant (Perlmutter et al., 2019; Gao et al., 2019).

## 3  RELAXED GEOMETRIC SCATTERING CONSTRUCTION TO ALLOW TRAINING

The geometric scattering construction, described in Sec. 2, can be seen as a particular GNN with handcrafted layers, rather than learned ones. This provides a solid mathematical framework for understanding the encoding of geometric information in GNNs, as shown in Perlmutter et al. (2019), while also providing effective unsupervised graph representation learning for data exploration, which also has some advantages even in supervised learning task, as shown in Gao et al. (2019). While the handcrafted design in Perlmutter et al. (2019); Gao et al. (2019) is not a priori amenable to task-driven tuning provided by end-to-end GNN training, we note that the cascade in Eq. 3 does

conform to a neural network architecture suitable for backpropagation. Therefore, in this section, we show how and under what conditions a relaxation of the laziness of the random walk and the selection of the scales preserves some of the useful mathematical properties established in Perlmutter et al. (2019). We then establish in section 5 the empirical benefits of learning the diffusion scales over a purely handcrafted design.

We first note that the construction of the diffusion matrix $\boldsymbol{P}$ that forms the lowpass filter used in the fixed scattering construction can be relaxed to encode adaptive laziness by setting $\boldsymbol{P}_\alpha := \alpha \boldsymbol{I}_n + (1 - \alpha)\boldsymbol{W}\boldsymbol{D}^{-1}$. Where $\alpha \in [1/2, 1)$ controls the reluctance of the random walk to transition from one node to another. $\alpha = 1/2$ gives an equal probability to stay in the same node as to transition to one of its neighbors. At this point, we note that one difference between the diffusion lowpass filter here and the one typically used in GCN and its variation is the symmetrization applied in Kipf & Welling (2016). However, Perlmutter et al. (2019) established that for the original construction, this is only a technical difference since $\boldsymbol{P}$ can be regarded as self-adjoint under an appropriate measure which encodes degree variations in the graph. This is then used to generate a Hilbert space $L^2(\mathcal{G}, \boldsymbol{D}^{-1/2})$ of graph signals with inner product $\langle \boldsymbol{x}, \boldsymbol{y} \rangle_{\boldsymbol{D}^{-1/2}} := \langle \boldsymbol{D}^{-1/2}\boldsymbol{x}, \boldsymbol{D}^{-1/2}\boldsymbol{y} \rangle$. The following lemma shows that a similar property is retained for our adaptive lowpass filter $\boldsymbol{P}_\alpha$.

**Lemma 1.** *The matrix $\boldsymbol{P}_\alpha$ is self-adjoint on the Hilbert space $L^2(\mathcal{G}, \boldsymbol{D}^{-1/2})$ from Perlmutter et al. (2019).*

We note that the self-adjointness shown here is interesting, as it links models that use symmetric and asymmetric versions of the Laplacian or adjacency matrix. Namely, Lemma 1 shows that the diffusion matrix $\boldsymbol{P}$ (which is column normalized but not row normalized) is self-adjoint, as an operator, and can thus be considered as "symmetric" in a suitable inner product space, thus establishing a theoretical link between these design choices.

As a second relaxation, we propose to replace the handcrafted dyadic scales in Eq. 1 with an adaptive monotonic sequence of integer diffusion time scales $0 < t_1 < \cdots < t_J$, which can be selected or tuned via training. Then, an adaptive filter bank is constructed as $\mathcal{W}'_J := \{\boldsymbol{\Psi}'_j, \boldsymbol{\Phi}'_J\}_{j=0}^{J-1}$, with

$$
\begin{aligned}
\boldsymbol{\Phi}'_J &:= \boldsymbol{P}_\alpha^{t_J}, \\
\boldsymbol{\Psi}'_0 &:= \boldsymbol{I}_n - \boldsymbol{P}_\alpha^{t_1}, \\
\boldsymbol{\Psi}'_j &:= \boldsymbol{P}_\alpha^{t_j} - \boldsymbol{P}_\alpha^{t_{j+1}}, \quad 1 \le j \le J - 1.
\end{aligned}
\tag{4}
$$

The following theorem shows that for any selection of scales, the relaxed construction of $\mathcal{W}'_J$ constructs a nonexpansive frame, similar to the result from Perlmutter et al. (2019) shown for the original handcrafted construction.

**Theorem 1.** *There exist a constant $C > 0$ that only depends on $t_1$ and $t_J$ such that for all $\boldsymbol{x} \in L^2(\mathcal{G}, \boldsymbol{D}^{-1/2})$,*

$$
C\|\boldsymbol{x}\|_{\boldsymbol{D}^{-1/2}}^2 \leqslant \|\boldsymbol{\Phi}'_J\boldsymbol{x}\|_{\boldsymbol{D}^{-1/2}}^2 + \sum_{j=0}^J \|\boldsymbol{\Psi}'_j\boldsymbol{x}\|_{\boldsymbol{D}^{-1/2}}^2 \leqslant \|\boldsymbol{x}\|_{\boldsymbol{D}^{-1/2}}^2,
$$

*where the norm considered here is the one induced by the space $L^2(\mathcal{G}, \boldsymbol{D}^{-1/2})$.*

Intuitively, the upper (i.e., nonexpansive) frame bound implies stability in the sense that small perturbations in the input graph signal will only result in small perturbations in the representation extracted by the constructed filter bank. Further, the lower frame bound ensures certain energy preservation by the constructed filter bank, thus indicating the nonexpansiveness is not implemented in a trivial fashion (e.g., by constant features independent of input signal).

In the next section we leverage the two relaxations described here to design a neural network architecture for learning the configuration $\alpha, t_1, \ldots, t_J$ of this relaxed construction via backpropagation through the resulting scattering filter cascade. The following theorem establishes that for any such configuration, extracted from $\mathcal{W}'_J$ via Eqs. 2-3, is permutation equivariant at the node-level and permutation invariant at the graph level. This guarantees that the extracted (in this case learned) features indeed encode intrinsic graph geometry rather than a priori indexation.

**Theorem 2.** *Let $\boldsymbol{U}'_p$ and $\boldsymbol{S}'_{p,q}$ be defined as in Eq. 2 and 3 (correspondingly), with the filters from $\mathcal{W}'_J$ with an arbitrary configuration $0 < \alpha < 1$, $0 < t_1 < \cdots < t_J$. Then, for any permutation $\Pi$*

over the nodes of $\mathcal{G}$, and any graph signal $\boldsymbol{x} \in L^2(\mathcal{G}, \boldsymbol{D}^{-1/2})$

$$\boldsymbol{U}'_p \Pi \boldsymbol{x} = \Pi \boldsymbol{U}'_p \boldsymbol{x} \quad \text{and} \quad \boldsymbol{S}'_{p,q} \Pi \boldsymbol{x} = \boldsymbol{S}'_{p,q} \boldsymbol{x} \qquad p \in \cup_{m \in \mathbb{N}} \mathbb{N}_0^m, q \in \mathbb{N}$$

*where geometric scattering implicitly considers here the node ordering supporting its input signal.*

We note that the results in Lemma 1 and Theorems 1-2, as well as their proofs, closely follow the theoretical framework proposed by Perlmutter et al. (2019). We carefully account here for the relaxed learned configuration, which replaces the originally handcrafted configuration there. For completeness, the adjusted proofs appear in Sec. A of the Appendix.

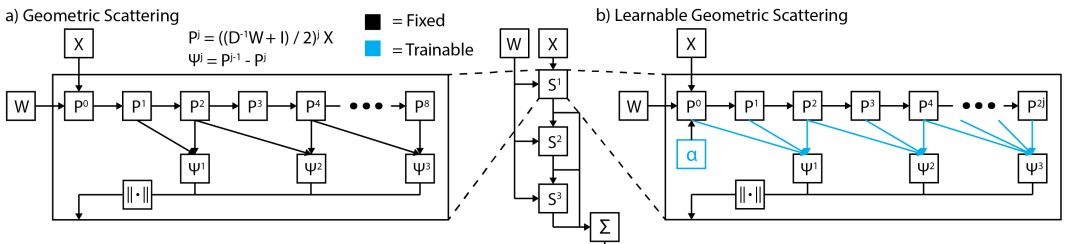

Figure 1: LEGSNet learns to select the appropriate scattering scales from the data.

# 4 LEARNABLE GEOMETRIC SCATTERING NETWORK ARCHITECTURE

In order to implement the relaxed geometric scattering construction (Sec. 3) via a trainable neural network, throughout this section, we consider an input graph signal $\boldsymbol{x} \in \mathbb{R}^n$ or, equivalently, a collection of graph signals $\boldsymbol{X} \in \mathbb{R}^{n \times N_{\ell-1}}$. The propagation of these signals can be divided into three major modules. First, a diffusion module implements the Markov process that forms the basis of the filter bank and transform, while allowing learning of the laziness parameter $\alpha$. Then, a scattering module implements the filters and the corresponding cascade, while allowing the learning of the scales $t_1, \ldots, t_J$. Finally, the aggregation module collects the extracted features to provide a graph and produces the task-dependent output.

**Building a diffusion process.** We build a set of $m \in \mathbb{N}$ subsequent diffusion steps of the signal $\boldsymbol{x}$ by iteratively multiplying the diffusion matrix $\boldsymbol{P}_\alpha$ to the left of the signal, resulting in

$$\left[ \boldsymbol{P}_\alpha \boldsymbol{x}, \boldsymbol{P}_\alpha^2 \boldsymbol{x}, \boldsymbol{P}_\alpha^3 \boldsymbol{x}, \ldots, \boldsymbol{P}_\alpha^m \boldsymbol{x} \right],$$

Since $\boldsymbol{P}_\alpha$ is often sparse, for efficiency reasons these filter responses are implemented via an RNN structure consisting of $m$ RNN modules. Each module propagates the incoming hidden state $\boldsymbol{h}_{t-1}, t = 1, \ldots, m$ with $\boldsymbol{P}_\alpha$ with the readout $\boldsymbol{o}_t$ equal to the produced hidden state,

$$\boldsymbol{h}_t \coloneqq \boldsymbol{P}_\alpha \boldsymbol{h}_{t-1}, \quad \boldsymbol{o}_t \coloneqq \boldsymbol{h}_t.$$

Our architecture and theory enable the implementation of either trainable or nontrainable $\alpha$, which we believe will be useful for future work as indicated, for example, in Gao & Ji (2019). However, in the applications considered here (see Sec. 5), we find that training $\alpha$ made training unstable and did not improve performance. Therefore, for simplicity, we leave it fixed as $\alpha = 1/2$ for the remainder of this work. In this case, the RNN portion of the network contains no trainable parameters, thus speeding up the computation, but still enables a convenient gradient flow back to the model input.

**Learning diffusion filter bank.** Next, we consider the selection of $J \leq m$ diffusion scales for the relaxed filter bank construction with the wavelets defined according to Eq. 5. We found this was the most influential part of the architecture. We experimented with methods of increasing flexibility:

1. Selection of $\{t_j\}_{j=1}^{J-1}$ as dyadic scales (as in Sec. 2 and Eq. 1), fixed for all datasets (LEGS-FIXED),

2. Selection of each $t_j$ using softmax and sorting by $j$, learnable per model (LEGS-FCN and LEGS-RBF, depending on output layer explained below).

For the softmax selection, we use a selection matrix $\boldsymbol{F} \in \mathbb{R}^{J \times m}$, where each row $\boldsymbol{F}_{(j,\cdot)}, j = 1, \ldots, J$ is dedicated to identifying the diffusion scale of the wavelet $\boldsymbol{P}_\alpha^{t_j}$ via a one-hot encoding. This is achieved by setting

$$\boldsymbol{F} := \operatorname{softmax}(\boldsymbol{\Theta}) = [\operatorname{softmax}(\boldsymbol{\theta}_1), \operatorname{softmax}(\boldsymbol{\theta}_2), \ldots, \operatorname{softmax}(\boldsymbol{\theta}_J)]^T$$

where $\boldsymbol{\theta}_j \in \mathbb{R}^m$ constitute the rows of the trainable weight matrix $\boldsymbol{\Theta}$. While this construction may not strictly guarantee an exact one-hot encoding, we assume that the softmax activations yield a sufficient approximation. Further, without loss of generality, we assume that the rows of $\boldsymbol{F}$ are ordered according to the position of the leading "one" activated in every row. In practice, this can be easily enforced by reordering the rows. We now construct the filter bank $\widetilde{\mathcal{W}}_{\boldsymbol{F}} := \{\widetilde{\boldsymbol{\Psi}}_j, \widetilde{\boldsymbol{\Phi}}_J\}_{j=0}^{J-1}$ with the filters

$$
\begin{aligned}
\widetilde{\boldsymbol{\Phi}}_J \boldsymbol{x} &= \sum\nolimits_{t=1}^m \boldsymbol{F}_{(J,t)} \boldsymbol{P}_\alpha^t \boldsymbol{x}, \\
\widetilde{\boldsymbol{\Psi}}_0 \boldsymbol{x} &= \boldsymbol{I}_n - \sum\nolimits_{t=1}^m \boldsymbol{F}_{(1,t)} \boldsymbol{P}_\alpha^t \boldsymbol{x} \\
\widetilde{\boldsymbol{\Psi}}_j \boldsymbol{x} &= \sum\nolimits_{t=1}^m \left[ \boldsymbol{F}_{(j,t)} \boldsymbol{P}_\alpha^t \boldsymbol{x} - \boldsymbol{F}_{j+1,t} \boldsymbol{P}_\alpha^t \boldsymbol{x} \right] \qquad 1 \le j \le J-1
\end{aligned}
\tag{5}
$$

matching and implementing the construction of $\mathcal{W}_J'$ from Eq. 4.

**Aggregating and classifying scattering features.** While many approaches may be applied to aggregate node-level features into graph-level features such as max, mean, sum pooling, and the more powerful TopK (Gao & Ji, 2019) or attention pooling (Veličković et al., 2018), we follow the statistical-moment aggregation explained in Secs. 2-3 (motivated by Gao et al., 2019; Perlmutter et al., 2019) and leave exploration of other pooling methods to future work. As shown in Gao et al. (2019) on graph classification, this aggregation works particularly well in conjunction with support vector machines (SVMs) based on the radial basis function (RBF) kernel.

Here, we consider two configurations for the task-dependent output layer of the network, either using a small neural network with two fully connected layers, which we denote LEGS-FCN, or using a modified RBF network (Broomhead & Lowe, 1988), which we denote LEGS-RBF, to produce the final classification. The latter configuration more accurately processes scattering features as shown in Table 2. Our RBF network works by first initializing a fixed number of movable anchor points. Then, for every point, new features are calculated based on the radial distances to these anchor points. In previous work on radial basis networks these anchor points were initialized independent of the data. We found that this led to training issues if the range of the data was not similar to the initialization of the centers. Instead, we first use a batch normalization layer to constrain the scale of the features and then pick anchors randomly from the initial features of the first pass through our data. This gives an RBF-kernel network with anchors that are always in the range of the data. Our RBF layer is then $\operatorname{RBF}(\boldsymbol{x}) = \phi(\|\operatorname{BatchNorm}(\boldsymbol{x}) - \boldsymbol{c}\|)$ with $\quad \phi(\boldsymbol{x}) = e^{-\|\boldsymbol{x}\|^2}$.

## 5 EMPIRICAL RESULTS

Here we show results of LEGSNet on whole graph classification and graph regression tasks, that arise in a variety of contexts, with emphasis on the more complex biochemic datasets. We use biochemical graph datasets as they represent a new challenge in the field of graph learning. Unlike

Table 1: Dataset statistics, diameter, nodes, edges, and clustering coefficient averaged over graphs.

|  | # Graphs | # Classes | Diameter | Nodes | Edges | Clust. Coeff |
|---|---|---|---|---|---|---|
| DD | 1178 | 2 | 19.81 | 284.32 | 715.66 | 0.48 |
| ENZYMES | 600 | 6 | 10.92 | 32.63 | 62.14 | 0.45 |
| MUTAG | 188 | 2 | 8.22 | 17.93 | 19.79 | 0.00 |
| NCI1 | 4110 | 2 | 13.33 | 29.87 | 32.30 | 0.00 |
| NCI109 | 4127 | 2 | 13.14 | 29.68 | 32.13 | 0.00 |
| PROTEINS | 1113 | 2 | 11.62 | 39.06 | 72.82 | 0.51 |
| PTC | 344 | 2 | 7.52 | 14.29 | 14.69 | 0.01 |

other types of data, these datasets do not exhibit the small-world structure of social datasets and may have large graph diameters for their size. Further, the connectivity patterns of biomolecules are very irregular due to 3D folding and long range connections, and thus ordinary local node aggregation methods may miss such connectivity differences.

## 5.1 Whole Graph Classification

We perform whole graph classification by using eccentricity and clustering coefficient as node features as is done in Gao et al. (2019). We compare against graph convolutional networks (GCN) (Kipf & Welling, 2016), GraphSAGE (Hamilton et al., 2017), graph attention network (GAT) (Veličković et al., 2018), graph isomorphism network (GIN) (Xu et al., 2019), Snowball network (Luan et al., 2019), and fixed geometric scattering with a support vector machine classifier (GS-SVM) as in Gao et al. (2019), and a baseline which is a 2-layer neural network on the features averaged across nodes (disregarding graph structure). These comparisons are meant to inform when including learnable graph scattering features are helpful in extracting whole graph features. Specifically, we are interested in the types of graph datasets where existing graph neural network performance can be improved upon with scattering features. We evaluate these methods across 7 benchmark biochemical datasets: DD, ENZYMES, MUTAG, NCI1, NCI109, PROTEINS, and PTC where the goal is to classify between two or more classes of compounds with hundreds to thousands of graphs and tens to hundreds of nodes (See Table 1). For completeness we also show results on six social network datasets in Table S2. For more specific information on individual datasets see Appendix B. We use 10-fold cross validation on all models which is elaborated on in Appendix C. For an ensembling comparison to Scattering-GCN (Min et al., 2020) see Appendix D.

Table 2: Mean $\pm$ standard deviation test set accuracy on biochemical datasets. Time limit expired (TLE) individual denotes models that did not finish in 10 hours.

| | DD | ENZYMES | MUTAG | NCI1 | NCI109 | PROTEINS | PTC |
|---|---|---|---|---|---|---|---|
| LEGS-RBF | $72.58 \pm 3.35$ | $36.33 \pm 4.50$ | $33.51 \pm 4.34$ | $\mathbf{74.26 \pm 1.53}$ | $\mathbf{72.47 \pm 2.11}$ | $70.89 \pm 3.91$ | $\mathbf{57.26 \pm 5.54}$ |
| LEGS-FCN | $72.07 \pm 2.37$ | $\mathbf{38.50 \pm 8.18}$ | $82.98 \pm 9.85$ | $70.83 \pm 2.65$ | $70.17 \pm 1.46$ | $71.06 \pm 3.17$ | $56.92 \pm 9.36$ |
| LEGS-FIXED | $69.09 \pm 4.82$ | $32.33 \pm 5.04$ | $81.84 \pm 11.24$ | $71.24 \pm 1.63$ | $69.25 \pm 1.75$ | $67.30 \pm 2.94$ | $54.31 \pm 6.92$ |
| GCN | $67.82 \pm 3.81$ | $31.33 \pm 6.89$ | $79.30 \pm 9.66$ | $60.80 \pm 4.26$ | $61.30 \pm 2.99$ | $74.03 \pm 3.20$ | $56.34 \pm 10.29$ |
| GraphSAGE | $66.37 \pm 4.45$ | $15.83 \pm 9.10$ | $81.43 \pm 11.64$ | $57.54 \pm 3.33$ | $55.15 \pm 2.58$ | $71.87 \pm 3.50$ | $55.22 \pm 9.13$ |
| GAT | $68.50 \pm 3.62$ | $25.83 \pm 4.73$ | $79.85 \pm 9.44$ | $62.19 \pm 2.18$ | $61.28 \pm 2.24$ | $73.22 \pm 3.55$ | $55.50 \pm 6.90$ |
| GIN | $42.37 \pm 4.32$ | $36.83 \pm 4.81$ | $83.57 \pm 9.68$ | $66.67 \pm 2.90$ | $65.23 \pm 1.82$ | $\mathbf{75.02 \pm 4.55}$ | $55.82 \pm 8.07$ |
| Snowball | TLE | $18.00 \pm 1.89$ | $50.56 \pm 20.87$ | $48.56 \pm 2.92$ | $50.86 \pm 2.65$ | $39.36 \pm 4.29$ | $50.84 \pm 9.32$ |
| GS-SVM | $72.66 \pm 4.94$ | $27.33 \pm 5.10$ | $\mathbf{85.09 \pm 7.44}$ | $69.68 \pm 2.38$ | $68.55 \pm 2.06$ | $70.98 \pm 2.67$ | $56.96 \pm 7.09$ |
| Baseline | $\mathbf{75.98 \pm 2.81}$ | $20.50 \pm 5.99$ | $79.80 \pm 9.92$ | $56.69 \pm 3.07$ | $57.38 \pm 2.20$ | $73.22 \pm 3.76$ | $56.71 \pm 5.54$ |

**LEGS outperforms on biological datasets.** A somewhat less explored domain for GNNs is in biochemical graphs that represent molecules and tend to be overall smaller and less connected (see Tables 1 and S1) than social networks. In particular we find that LEGSNet outperforms other methods by a significant margin on biochemical datasets with relatively small but high diameter graphs (NCI1, NCI109, ENZYMES, PTC), as shown in Table 2. On extremely small graphs we find that GS-SVM performs best, which is expected as other methods with more parameters can easily overfit the data. We reason that the performance increases exhibited by LEGSNet, and to a lesser extent GS-SVM, on these chemical and biological benchmarks is due the ability of geometric scattering to compute complex connectivity features via its multiscale diffusion wavelets. Thus, methods that rely on a scattering construction would in general perform better, with the flexibility and trainability LEGSNet giving it an edge on most tasks.

**LEGS performs consistently on social network datasets.** On the social network datasets LEGSNet performs consistently well, although its benefits here are not as clear as in the biochemical datasets. Ignoring the fixed scattering transform GS-SVM, which was tuned in Gao et al. (2019) with a focus on these particular social network datasets, a version of LEGSNet is best on three out of the six social datasets and second best on the other three. Since the advantages are clearer in the biochemical domain, we focus on this in the remainder of this section. However, for completeness, we provide results on social network datasets in Table S2, and leave further discussion to Appendix B.1.

**LEGS preserves enzyme exchange preferences while increasing performance.** One advantage of geometric scattering over other graph embedding techniques lies in the rich information present within the scattering feature space. This was demonstrated in Gao et al. (2019) where it was shown that the embeddings created through fixed geometric scattering can be used to accurately infer inter-graph relationships. Scattering features of enzyme graphs within the ENZYMES dataset (Borgwardt et al., 2005) possessed sufficient

global information to recreate the enzyme class exchange preferences observed empirically by Cuesta et al. (2015), using only linear methods of analysis, and despite working with a much smaller and artificially balanced dataset. We demonstrate here that LEGSNet retains similar descriptive capabilities, as shown in Figure 2 via chord diagrams where each exchange preference between enzyme classes (estimated as suggested in Gao et al., 2019) is represented as a ribbon of the corresponding size. Our results here (and in Table S5, which provides complementary quantitative comparison) show that, with relaxations on the scattering parameters, LEGS-FCN achieves better classification accuracy than both LEGS-FIXED and GCN (see Table 1) while also retaining a more descriptive embedding that maintains the global structure of relations between enzyme classes. We ran two varieties of LEGSNet on the EN-ZYMES dataset: LEGS-FIXED and LEGS-FCN, which allows the diffusion scales to be learned. For comparison, we also ran a standard GCN whose graph embeddings were obtained via mean pooling. To infer enzyme ex-

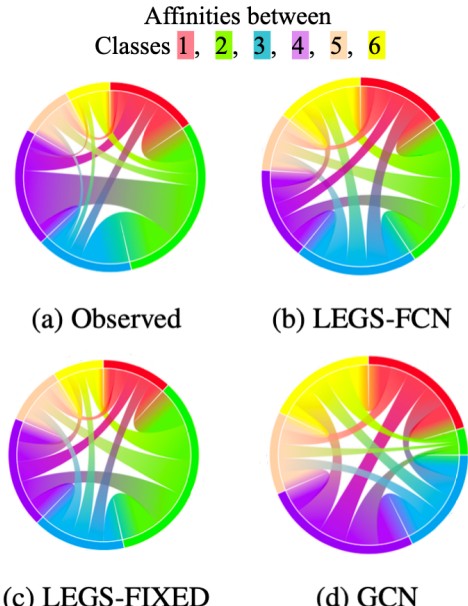

Affinities between
Classes 1, 2, 3, 4, 5, 6

(a) Observed      (b) LEGS-FCN

(c) LEGS-FIXED      (d) GCN

Figure 2: Enzyme class exchange preferences empirically observed in Cuesta et al. (2015), and estimated from LEGS and GCN embeddings.

change preferences from their embeddings, we followed Gao et al. (2019) in defining the distance from an enzyme $e$ to the enzyme class $\mathrm{EC}_j$ as $\mathrm{dist}(e, \mathrm{EC}_j) := \|v_e - \mathrm{proj}_{C_j}(v_e)\|$, where $v_i$ is the embedding of $e$, and $C_j$ is the PCA subspace of the enzyme feature vectors within $\mathrm{EC}_j$. The distance between the enzyme classes $\mathrm{EC}_i$ and $\mathrm{EC}_j$ is the average of the individual distances, $\mathrm{mean}\{\mathrm{dist}(e, \mathrm{EC}_j) : e \in \mathrm{EC}_i\}$. From here, the affinity between two enzyme classes is computed as $\mathrm{pref}(\mathrm{EC}_i, \mathrm{EC}_j) = w_i / \min(\frac{D_{i,i}}{D_{i,j}}, \frac{D_{j,j}}{D_{j,i}})$, where $w_i$ is the percentage of enzymes in class $i$ which are closer to another class than their own, and $D_{i,j}$ is the distance between $\mathrm{EC}_i$ and $\mathrm{EC}_j$.

**Robustness to reduced training set size.** We remark that similar to the robustness shown in (Gao et al., 2019) for handcrafted scattering, LEGSNet is able to maintain accuracy even when the training set size is shrunk to as low as 20% of the dataset, with a median decrease of 4.7% accuracy as when 80% of the data is used for training, as discussed in the supplement (see Table S3).

## 5.2 GRAPH REGRESSION

We next evaluate learnable scattering on two graph regression tasks, the QM9 (Gilmer et al., 2017; Wu et al., 2018) graph regression dataset, and a new task from the critical assessment of structure prediction (CASP) challenge (Moult et al., 2018). On the CASP task, the main objective is to score protein structure prediction/simulation models in terms of the discrepancy between their predicted structure and the actual structure of the protein (which is known a priori). The accuracy of such 3D structure predictions are evaluated using a variety of met-

Table 3: Train and test set mean squared error on CASP GDT regression task over three seeds.

| $(\mu \pm \sigma)$ | Train MSE | Test MSE |
|---|---|---|
| LEGS-FCN | **134.34 ± 8.62** | **144.14 ± 15.48** |
| LEGS-RBF | 140.46 ± 9.76 | 152.59 ± 14.56 |
| LEGS-FIXED | 136.84 ± 15.57 | 160.03 ± 1.81 |
| GCN | 289.33 ± 15.75 | 303.52 ± 18.90 |
| GraphSAGE | 221.14 ± 42.56 | 219.44 ± 34.84 |
| GIN | 221.14 ± 42.56 | 219.44 ± 34.84 |
| Baseline | 393.78 ± 4.02 | 402.21 ± 21.45 |

rics, but we focus on the global distance test (GDT) score (Modi et al., 2016). The GDT score measures the similarity between tertiary structures of two proteins with amino-acid correspondence. A higher score means two structures are more similar. For a set of predicted 3D structures for a protein, we would like to score their quality as quantified by the GDT score.

For this task we use the CASP12 dataset (Moult et al., 2018) and preprocess the data similarly to Ingraham et al. (2019), creating a KNN graph between proteins based on the 3D coordinates of each amino acid. From this KNN graph we regress against the GDT score. We evaluate on 12 proteins from the CASP12 dataset and choose random (but consistent) splits with 80% train, 10% validation, and 10% test data out of 4000 total structures. We are only concerned with structure similarity so use no non-structural node features.

**LEGSNet outperforms on all CASP targets**
Across all CASP targets we find that LEGSNet significantly outperforms GNN and baseline methods (See Table S4). This performance improvement is particularly stark on the easiest structures (measured by average GDT) but is consistent across all structures. In Figure 3 we show the relationship between percent improvement of LEGSNet over the GCN model and the average GDT score across the target structures. We draw attention to target t0879, where LEGSNet shows the greatest improvement over other methods. This target has long range dependencies (Ovchinnikov et al., 2018) as it exhibits metal coupling (Li et al., 2015)

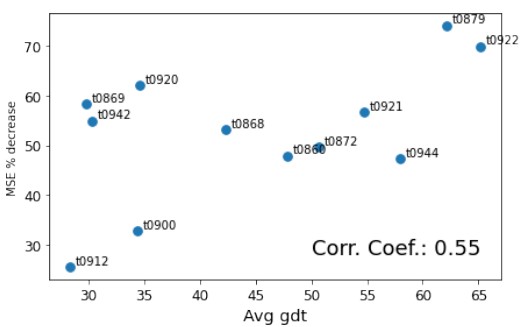

Figure 3: CASP dataset LEGS-FCN % improvement over GCN in MSE of GDT prediction vs. Average GDT score.

creating long range connections over the sequence. Since other methods are unable to model these long range connections LEGSNet is particularly important on these more difficult to model targets.

**LEGSNet outperforms on the QM9 dataset** We evaluate the performance of LEGSNet on the quantum chemistry dataset QM9 (Gilmer et al., 2017; Wu et al., 2018), which consists of 130,000 molecules with ~18 nodes per molecule. We use the node features from Gilmer et al. (2017), with the addition of eccentricity and clustering coefficient features, and ignore the edge features. We whiten all targets to have zero mean and unit standard deviation. We train each network against all 19 targets and evaluate the mean squared error on the test set with mean and std. over four runs. We find that learning the scales improves the overall MSE, and particularly improves the results over difficult targets (see Table 4 for overall results and Table S7 for results by target). Indeed, on more difficult targets (i.e., those with large test error) LEGS-FCN is able to

Table 4: Mean ± std. over four runs of mean squared error over 19 targets for the QM9 dataset, lower is better.

| $(\mu \pm \sigma)$ | Test MSE |
|---|---|
| LEGS-FCN | **0.216 ± 0.009** |
| LEGS-FIXED | 0.228 ± 0.019 |
| GraphSAGE | 0.524 ± 0.224 |
| GCN | 0.417 ± 0.061 |
| GIN | 0.247 ± 0.037 |
| Baseline | 0.533 ± 0.041 |

perform better, where on easy targets GIN is the best. Overall, scattering features offer a robust signal over many targets, and while perhaps less flexible (by construction), they achieve good average performance with significantly fewer parameters.

## 6 CONCLUSION

In this work we have established a relaxation from fixed geometric scattering with strong guarantees to a more flexible network with better performance by learning data dependent scales. Allowing the network to choose data-driven diffusion scales leads to improved performance particularly on biochemical datasets, while keeping strong guarantees on extracted features. This parameterization has advantages in representing long range connections with a small number of weights, which are necessary in complex biochemical data. This also opens the possibility to provide additional relaxation to enable node-specific or graph-specific tuning via attention mechanisms, which we regard as an exciting future direction, but out of scope for the current work.

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

APPENDIX

## A  PROOFS FOR SECTION 3

### A.1  PROOF OF LEMMA 1

Let $M_\alpha = D^{-1/2} P_\alpha D^{1/2}$ then it can be verified that $M_\alpha$ is a symmetric conjugate of $P_\alpha$, and by construction is self-adjoint with respect to the standard inner product of $L^2(\mathcal{G})$. Let $x, y \in L^2(\mathcal{G}, D^{-1/2})$ then we have

$$
\begin{aligned}
\langle P_\alpha x, y \rangle_{D^{-1/2}} &= \langle D^{-1/2} P_\alpha x, D^{-1/2} y \rangle \\
&= \langle D^{-1/2} D^{1/2} M_\alpha D^{-1/2} x, D^{-1/2} y \rangle \\
&= \langle M_\alpha D^{-1/2} x, D^{-1/2} y \rangle \\
&= \langle D^{-1/2} x, M_\alpha D^{-1/2} y \rangle \\
&= \langle D^{-1/2} x, D^{-1/2} D^{1/2} M_\alpha D^{-1/2} y \rangle \\
&= \langle D^{-1/2} x, D^{-1/2} P_\alpha y \rangle \\
&= \langle x, P_\alpha y \rangle_{D^{-1/2}},
\end{aligned}
$$

which gives the result of the lemma. $\qquad \square$

### A.2  PROOF OF THEOREM 1

As shown in the previous proof (Sec. A.1), $P_\alpha$ has a symmetric conjugate $M_\alpha$. Given the eigendecomposition $M_\alpha = Q \Lambda Q^T$, we can write $P_\alpha^t = D^{1/2} Q \Lambda^t Q^T D^{-1/2}$, giving the eigendecomposition of the propagated diffusion matrices. Furthermore, it can be verified that the eigenvalues on the diagonal of $\Lambda$ are nonnegative. Briefly, this results from graph Laplacian eigenvalues being within the range $[0, 1]$, which means those of $WD^{-1}$ are in $[-1, 1]$, which combined with $1/2 \le \alpha \le 1$ result in $\lambda_i := [\Lambda]_{ii} \in [0, 1]$ for every $j$. Next, given this decomposition we can write:

$$
\begin{aligned}
\Phi'_J &= D^{1/2} Q \Lambda^{t_J} Q^T D^{-1/2}, \\
\Psi'_j &= D^{1/2} Q (\Lambda^{t_j} - \Lambda^{t_{j+1}}) Q^T D^{-1/2}, \quad 0 \le j \le J-1.
\end{aligned}
$$

where we set $t_0 = 0$ to simplify notations. Then, we have:

$$
\begin{aligned}
\| \Phi'_J x \|_{D^{-1/2}}^2 &= \langle \Phi'_J x, \Phi'_J x \rangle_{D^{-1/2}} \\
&= \langle D^{-1/2} D^{1/2} Q \Lambda^{t_J} Q^T D^{-1/2} x, D^{-1/2} D^{1/2} Q \Lambda^{t_J} Q^T D^{-1/2} x \rangle \\
&= x^T D^{-1/2} Q \Lambda^{t_J} Q^T Q \Lambda^{t_J} Q^T D^{-1/2} x = (x^T D^{-1/2} Q \Lambda^{t_J})(\Lambda^{t_J} Q^T D^{-1/2} x) \\
&= \| \Lambda^{t_J} Q^T D^{-1/2} x \|_2^2
\end{aligned}
$$

Further, since $Q$ is orthogonal (as it is constructed from an eigenbasis of a symmetric matrix), if we consider a change of variable to $y = Q^T D^{-1/2} x$, we have $\| x \|_{D^{-1/2}}^2 = \| D^{-1/2} x \|_2^2 = \| y \|_2^2$ while $\| \Phi'_J x \|_{D^{-1/2}}^2 = \| \Lambda^{t_J} y \|_2^2$. Similarly, we can also reformulate the operation of other filters in terms of diagonal matrices applied to $y$ as $\mathcal{W}'_J$ as $\| \Psi'_j x \|_{D^{-1/2}}^2 = \| (\Lambda^{t_j} - \Lambda^{t_{j+1}}) y \|_2^2$.

Given the reformulation in terms of $y$ and standard $L^2(\mathcal{G})$, we can now write

$$
\| \Lambda^{t_J} y \|_2^2 + \sum_{j=0}^{J-1} \| (\Lambda^{t_j} - \Lambda^{t_{j+1}}) y \|_2^2 = \sum_{i=1}^{n} y_i^2 \cdot \left( \lambda^{2t_J} + \sum_{j=0}^{J-1} (\lambda_i^{t_j} - \lambda_i^{t_{j+1}})^2 \right).
$$

Then, since $0 \le \lambda_i \le 1$ and $0 = t_0 < t_1 < \cdots < t_J$ we have

$$
\lambda^{2t_J} + \sum_{j=0}^{J-1} (\lambda_i^{t_j} - \lambda_i^{t_{j+1}})^2 \le \left( \lambda^{t_J} + \sum_{j=0}^{J-1} \lambda_i^{t_j} - \lambda_i^{t_{j+1}} \right)^2 = \left( \lambda^{t_J} + \lambda_i^{t_0} - \lambda_i^{t_J} \right)^2 = 1,
$$

which yields the upper bound $\|\Lambda^{t_J}\boldsymbol{y}\|_2^2 + \sum_{j=0}^{J-1}\|(\Lambda^{t_j} - \Lambda^{t_{j+1}})\boldsymbol{y}\|_2^2 \leq \|\boldsymbol{y}\|_2^2$. On the other hand, since $t_1 > 0 = t_0$, then we also have

$$\lambda^{2t_J} + \sum_{j=0}^{J-1}(\lambda_i^{t_j} - \lambda_i^{t_{j+1}})^2 \geq \lambda^{2t_J} + (1 - \lambda_i^{t_1})^2$$

and therefore, by setting $C := \min_{0 \leq \xi \leq 1}(\xi^{2t_J} + (1 - \xi^{t_1})^2) > 0$, whose positivity is not difficult to verify, we get the lower bound $\|\Lambda^{t_J}\boldsymbol{y}\|_2^2 + \sum_{j=0}^{J-1}\|(\Lambda^{t_j} - \Lambda^{t_{j+1}})\boldsymbol{y}\|_2^2 \geq C\|\boldsymbol{y}\|_2^2$. Finally, applying the reverse change of variable to $\boldsymbol{x}$ and $L^2(\mathcal{G}, \boldsymbol{D}^{-1/2})$ yields the result of the theorem. $\qquad\square$

### A.3 Proof of Theorem 2

Denote the permutation group on $n$ elements as $S_n$, then for a permutation $\Pi \in S_n$ we let $\overline{\mathcal{G}} = \Pi(\mathcal{G})$ be the graph obtained by permuting the vertices of $\mathcal{G}$ with $\Pi$. The corresponding permutation operation on a graph signal $\boldsymbol{x} \in L^2(\mathcal{G}, \boldsymbol{D}^{-1/2})$ gives a signal $\Pi\boldsymbol{x} \in L^2(\overline{\mathcal{G}}, \boldsymbol{D}^{-1/2})$, which we implicitly considered in the statement of the theorem, without specifying these notations for simplicity. Rewriting the statement of the theorem more rigorously with the introduced notations, we aim to show that $\overline{\boldsymbol{U}}_p'\Pi\boldsymbol{x} = \Pi\boldsymbol{U}_p'\boldsymbol{x}$ and $\overline{\boldsymbol{S}}_{p,q}'\Pi\boldsymbol{x} = \boldsymbol{S}_{p,q}'\boldsymbol{x}$ under suitable conditions, where the operation $\boldsymbol{U}_p'$ from $\mathcal{G}$ on the permuted graph $\overline{\mathcal{G}}$ is denoted here by $\overline{\boldsymbol{U}}_p'$ and likewise for $\boldsymbol{S}_{p,q}'$ we have $\overline{\boldsymbol{S}}_{p,q}'$.

We start by showing $\boldsymbol{U}_p'$ is permutation equivariant. First, we notice that for any $\Psi_j$, $0 < j < J$ we have that $\overline{\Psi}_j\Pi\boldsymbol{x} = \Pi\Psi_j\boldsymbol{x}$, as for $1 \leq j \leq J - 1$

$$\begin{aligned}\overline{\Psi}_j\Pi\boldsymbol{x} &= (\Pi\boldsymbol{P}^{t_j}\Pi^T - \Pi\boldsymbol{P}^{t_{j+1}}\Pi^T)\Pi\boldsymbol{x} \\ &= \Pi(\boldsymbol{P}^{t_j} - \boldsymbol{P}^{t_{j+1}})\boldsymbol{x} \\ &= \Pi\Psi_j\boldsymbol{x}.\end{aligned}$$

Similar reasoning also holds for $j \in \{0, J\}$. Further, notice that for the element-wise nature of the absolute value nonlinearity yields $|\Pi\boldsymbol{x}| = \Pi|\boldsymbol{x}|$ for any permutation matrix $\Pi$. Using these two observations, it follows inductively that

$$\begin{aligned}\overline{\boldsymbol{U}}_p'\Pi\boldsymbol{x} &:= \boldsymbol{\Psi}_{j_m}'|\boldsymbol{\Psi}_{j_{m-1}}' \ldots |\boldsymbol{\Psi}_{j_2}'|\boldsymbol{\Psi}_{j_1}'\Pi\boldsymbol{x}||\ldots| \\ &= \boldsymbol{\Psi}_{j_m}'|\boldsymbol{\Psi}_{j_{m-1}}' \ldots |\boldsymbol{\Psi}_{j_2}'\Pi|\boldsymbol{\Psi}_{j_1}'\boldsymbol{x}||\ldots| \\ &\;\;\vdots \\ &= \Pi\boldsymbol{\Psi}_{j_m}'|\boldsymbol{\Psi}_{j_{m-1}}' \ldots |\boldsymbol{\Psi}_{j_2}'|\boldsymbol{\Psi}_{j_1}'\boldsymbol{x}||\ldots| \\ &= \Pi\boldsymbol{U}_p'\boldsymbol{x}.\end{aligned}$$

To show $\boldsymbol{S}_{p,q}'$ is permutation invariant, first notice that for any statistical moment $q > 0$, we have $|\Pi\boldsymbol{x}|^q = \Pi|\boldsymbol{x}|^q$ and further as sums are commutative, $\sum_j(\Pi\boldsymbol{x})_j = \sum_j \boldsymbol{x}_j$. We then have

$$\overline{\boldsymbol{S}}_{p,q}'\Pi\boldsymbol{x} = \sum_{i=1}^n |\overline{\boldsymbol{U}}_p'\Pi\boldsymbol{x}[v_i]|^q = \sum_{i=1}^n |\Pi\boldsymbol{U}_p'\boldsymbol{x}[v_i]|^q = \sum_{i=1}^n |\boldsymbol{U}_p'\boldsymbol{x}[v_i]|^q = \boldsymbol{S}_{p,q}'\boldsymbol{x},$$

which, together with the previous result, completes the proof of the theorem. $\qquad\square$

## B Datasets

In this section we further analyze individual datasets. Relating composition of the dataset as shown in Table S1 to the relative performance of our models as shown in Table S2.

**DD** Dobson & Doig (2003): Is a dataset extracted from the protein data bank (PDB) of 1178 high resolution proteins. The task is to distinguish between enzymes and non-enzymes. Since these are high resolution structures, these graphs are significantly larger than those found in our other biochemical datasets with a mean graph size of 284 nodes with the next largest biochemical dataset with a mean size of 39 nodes.

**ENZYMES**   Borgwardt et al. (2005): Is a dataset of 600 enzymes divided into 6 balanced classes of 100 enzymes each. As we analyzed in the main text, scattering features are better able to preserve the structure between classes. LEGS-FCN slightly relaxes this structure but improves accuracy from 32 to 39% over LEGS-FIXED.

**NCI1, NCI109**   Wale et al. (2008): Contains slight variants of 4100 chemical compounds encoded as graphs. Each compound is separated into one of two classes based on its activity against non-small cell lung cancer and ovarian cancer cell lines. Graphs in this dataset are 30 nodes with a similar number of edges. This makes for long graphs with high diameter.

**PROTEINS**   Borgwardt et al. (2005): Contains 1178 protein structures with the goal of classifying enzymes vs. non enzymes. GCN outperforms all other models on this dataset, however the Baseline model, where no structure is used also performs very similarly. This suggests that the graph structure within this dataset does not add much information over the structure encoded in the eccentricity and clustering coefficient.

**PTC**   Toivonen et al. (2003): Contains 344 chemical compound graphs divided into two classes based on whether or not they cause cancer in rats. This dataset is very difficult to classify without features however LEGS-RBF and LEGS-FCN are able to capture the long range connections slightly better than other methods.

**COLLAB**   Yanardag & Vishwanathan (2015): 5000 ego-networks of different researchers from high energy physics, condensed matter physics or astrophysics. The goal is to determine which field the research belongs to. The GraphSAGE model performs best on this dataset although the LEGS-RBF network performs nearly as well. Ego graphs have a very small average diameter. Thus shallow networks can perform quite well on them as is the case here.

**IMDB**   Yanardag & Vishwanathan (2015): For each graph nodes represent actresses/actors and there is an edge between them if they are in the same move. These graphs are also ego graphs around specific actors. IMDB-BINARY classifies between action and romance genres. IMDB-MULTI classifies between 3 classes. Somewhat surprisingly GS-SVM performs the best with other LEGS networks close behind. This could be due to oversmoothing on the part of GCN and GraphSAGE when the graphs are so small.

**REDDIT**   Yanardag & Vishwanathan (2015): Graphs in REDDIT-BINARY/MULTI-5K/MULTI-12K datasets each graph represents a discussion thread where nodes correspond to users and there is an edge between two nodes if one replied to the other's comment. The task is to identify which subreddit a given graph came from. On these datasets GCN outperforms other models.

**QM9**   Gilmer et al. (2017); Wu et al. (2018): Graphs in the QM9 dataset each represent chemicals with 18 atoms. Regression targets represent chemical properties of the molecules.

### B.1   PERFORMANCE OF LEGSNET ON SOCIAL NETWORK DATASETS

Table S2 shows that our model outperforms other GNNs on some biomedical benchmarks and that it performs comparably on social network datasets. Out of the six social network datasets, ignoring the fixed scattering model GS-SVM, which has been hand tuned with these datasets in mind, our model outperforms both GNN models on three of them, and is second best on the other three. This is at least comparable if not slightly superior performance. GraphSAGE does a bit better on Collab, but much worse on IMDB-Binary and Reddit-Binary. GCN does a bit better on Reddit-Multi, but worse on Collab, IMDB-Binary, and Reddit-Binary.

LEGSNet has significantly fewer parameters and achieves comparable or superior accuracy on common benchmarks. Even when our method shows comparable results, and definitely when it outperforms other GNNs, we believe that its smaller number of parameters could be useful in applications with limited compute or limited training examples.

Table S1: Dataset statistics, diameter, nodes, edges, clustering coefficient averaged over all graphs. Split into bio-chemical and social network types.

| | # Graphs | # Classes | Diameter | Nodes | Edges | Clust. Coeff |
|---|---|---|---|---|---|---|
| DD | 1178 | 2 | 19.81 | 284.32 | 715.66 | 0.48 |
| ENZYMES | 600 | 6 | 10.92 | 32.63 | 62.14 | 0.45 |
| MUTAG | 188 | 2 | 8.22 | 17.93 | 19.79 | 0.00 |
| NCI1 | 4110 | 2 | 13.33 | 29.87 | 32.30 | 0.00 |
| NCI109 | 4127 | 2 | 13.14 | 29.68 | 32.13 | 0.00 |
| PROTEINS | 1113 | 2 | 11.62 | 39.06 | 72.82 | 0.51 |
| PTC | 344 | 2 | 7.52 | 14.29 | 14.69 | 0.01 |
| COLLAB | 5000 | 3 | 1.86 | 74.49 | 2457.22 | 0.89 |
| IMDB-BINARY | 1000 | 2 | 1.86 | 19.77 | 96.53 | 0.95 |
| IMDB-MULTI | 1500 | 3 | 1.47 | 13.00 | 65.94 | 0.97 |
| REDDIT-BINARY | 2000 | 2 | 8.59 | 429.63 | 497.75 | 0.05 |
| REDDIT-MULTI-12K | 11929 | 11 | 9.53 | 391.41 | 456.89 | 0.03 |
| REDDIT-MULTI-5K | 4999 | 5 | 10.57 | 508.52 | 594.87 | 0.03 |

Table S2: Mean $\pm$ std. over 10 test sets on bio-chemical and social datasets.

| | LEGS-RBF | LEGS-FCN | LEGS-FIXED | GCN | GraphSAGE | GAT | GIN | GS-SVM | Baseline |
|---|---|---|---|---|---|---|---|---|---|
| DD | 72.58 ± 3.35 | 72.07 ± 2.37 | 69.09 ± 4.82 | 67.82 ± 3.81 | 66.37 ± 4.45 | 68.50 ± 3.62 | 42.37 ± 4.32 | 72.66 ± 4.94 | **75.98 ± 2.81** |
| ENZYMES | 36.33 ± 4.50 | **38.50 ± 8.18** | 32.33 ± 5.04 | 31.33 ± 6.89 | 15.83 ± 9.10 | 25.83 ± 4.73 | 36.83 ± 4.81 | 27.33 ± 5.10 | 20.50 ± 5.99 |
| MUTAG | 33.51 ± 4.34 | 82.98 ± 9.85 | 81.84 ± 11.24 | 79.30 ± 9.66 | 81.43 ± 11.64 | 79.85 ± 9.44 | 83.57 ± 9.68 | **85.09 ± 7.44** | 79.80 ± 9.92 |
| NCI1 | **74.26 ± 1.53** | 70.83 ± 2.65 | 71.24 ± 1.63 | 60.80 ± 4.26 | 57.54 ± 3.33 | 62.19 ± 2.18 | 66.67 ± 2.90 | 69.68 ± 2.38 | 56.69 ± 3.07 |
| NCI109 | **72.47 ± 2.11** | 70.17 ± 1.46 | 69.25 ± 1.75 | 61.30 ± 2.99 | 55.15 ± 2.58 | 61.28 ± 2.24 | 65.23 ± 1.82 | 68.55 ± 2.06 | 57.38 ± 2.20 |
| PROTEINS | 70.89 ± 3.91 | 71.06 ± 3.17 | 67.30 ± 2.94 | 74.03 ± 3.20 | 71.87 ± 3.50 | 73.22 ± 3.55 | **75.02 ± 4.55** | 70.98 ± 2.67 | 73.22 ± 3.76 |
| PTC | **57.26 ± 5.54** | 56.92 ± 9.36 | 54.31 ± 6.92 | 56.34 ± 10.29 | 55.22 ± 9.13 | 55.50 ± 6.90 | 55.82 ± 8.07 | 56.96 ± 7.09 | 56.71 ± 5.54 |
| COLLAB | 75.78 ± 1.95 | 75.40 ± 1.80 | 72.94 ± 1.70 | 73.80 ± 1.73 | **76.12 ± 1.58** | 72.88 ± 2.06 | 62.98 ± 3.92 | 74.54 ± 2.32 | 64.76 ± 2.63 |
| IMDB-BINARY | 64.90 ± 3.48 | 64.50 ± 3.50 | 64.30 ± 3.68 | 47.40 ± 6.24 | 46.40 ± 4.03 | 45.50 ± 3.14 | 64.20 ± 5.77 | **66.70 ± 3.53** | 47.20 ± 5.67 |
| IMDB-MULTI | 41.93 ± 3.01 | 40.13 ± 2.77 | 41.67 ± 3.19 | 39.33 ± 3.13 | 39.73 ± 3.45 | 39.73 ± 3.61 | 38.67 ± 3.93 | **42.13 ± 2.53** | 39.53 ± 3.63 |
| REDDIT-BINARY | **86.10 ± 2.92** | 78.15 ± 5.42 | 85.00 ± 1.93 | 81.60 ± 2.32 | 73.40 ± 4.38 | 73.35 ± 2.27 | 71.40 ± 6.98 | 85.15 ± 2.78 | 69.30 ± 5.08 |
| REDDIT-MULTI-12K | 38.47 ± 1.07 | 38.46 ± 1.31 | 39.74 ± 1.31 | **42.57 ± 0.90** | 32.17 ± 2.04 | 32.74 ± 0.75 | 24.45 ± 5.52 | 39.79 ± 1.11 | 22.07 ± 0.98 |
| REDDIT-MULTI-5K | 47.83 ± 2.61 | 46.97 ± 3.06 | 47.17 ± 2.93 | **52.79 ± 2.11** | 45.71 ± 2.88 | 44.03 ± 2.57 | 35.73 ± 8.35 | 48.79 ± 2.95 | 36.41 ± 1.80 |

## C  TRAINING DETAILS

We train all models for a maximum of 1000 epochs with an initial learning rate of $1e^{-4}$ using the ADAM optimizer (Kingma & Ba, 2015). We terminate training if validation loss does not improve for 100 epochs testing every 10 epochs. Our models are implemented with Pytorch Paszke et al. (2019) and Pytorch geometric. Models were run on a variety of hardware resources. For all models we use $q = 4$ normalized statistical moments for the node to graph level feature extraction and $m = 16$ diffusion scales in line with choices in Gao et al. (2019).

### C.1  CROSS VALIDATION PROCEDURE

For all datasets we use 10-fold cross validation with 80% training data 10% validation data and 10% test data for each model. We first split the data into 10 (roughly) equal partitions. For each model we take exactly one of the partitions to be the test set and one of the remaining nine to be the validation set. We then train the model on the remaining eight partitions using the cross-entropy loss on the validation for early stopping checking every ten epochs. For each test set, we use majority voting of the nine models trained with that test set. We then take the mean and standard deviation across these test set scores to average out any variability in the particular split chosen. This results in 900 models trained on every dataset. With mean and standard deviation over 10 ensembled models each with a separate test set.

## D  ENSEMBLING EVALUATION

Recent work by Min et al. (2020) combines the features from a fixed scattering transform with a GCN network, showing that this has empirical advantages in semi-supervised node classification, and theoretical representation advantages over a standard Kipf & Welling (2016) style GCN. We

Table S3: Mean $\pm$ std. over test set selection on cross-validated LEGS-RBF Net with reduced training set size.

| Train, Val, Test % | 80%, 10%, 10% | 70%, 10%, 20% | 40%, 10%, 50% | 20%, 10%, 70% |
|---|---|---|---|---|
| COLLAB | 75.78 ± 1.95 | 75.00 ± 1.83 | 74.00 ± 0.51 | 72.73 ± 0.59 |
| DD | 72.58 ± 3.35 | 70.88 ± 2.83 | 69.95 ± 1.85 | 69.43 ± 1.24 |
| ENZYMES | 36.33 ± 4.50 | 34.17 ± 3.77 | 29.83 ± 3.54 | 23.98 ± 3.32 |
| IMDB-BINARY | 64.90 ± 3.48 | 63.00 ± 2.03 | 63.30 ± 1.27 | 57.67 ± 6.04 |
| IMDB-MULTI | 41.93 ± 3.01 | 40.80 ± 1.79 | 41.80 ± 1.23 | 36.83 ± 3.31 |
| MUTAG | 33.51 ± 4.34 | 33.51 ± 1.14 | 33.52 ± 1.26 | 33.51 ± 0.77 |
| NCI1 | 74.26 ± 1.53 | 74.38 ± 1.38 | 72.07 ± 0.28 | 70.30 ± 0.72 |
| NCI109 | 72.47 ± 2.11 | 72.21 ± 0.92 | 70.44 ± 0.78 | 68.46 ± 0.96 |
| PROTIENS | 70.89 ± 3.91 | 69.27 ± 1.95 | 69.72 ± 0.27 | 68.96 ± 1.63 |
| PTC | 57.26 ± 5.54 | 57.83 ± 4.39 | 54.62 ± 3.21 | 55.45 ± 2.35 |
| REDDIT-BINARY | 86.10 ± 2.92 | 86.05 ± 2.51 | 85.15 ± 1.77 | 83.71 ± 0.97 |
| REDDIT-MULTI-12K | 38.47 ± 1.07 | 38.60 ± 0.52 | 37.55 ± 0.05 | 36.65 ± 0.50 |
| REDDIT-MULTI-5K | 47.83 ± 2.61 | 47.81 ± 1.32 | 46.73 ± 1.46 | 44.59 ± 1.02 |

Table S4: Test set mean squared error on CASP GDT regression task across targets over 3 non-overlapping test sets.

| | LEGS-RBF | LEGS-FCN | LEGS-FIXED | GCN | GraphSAGE | GIN | Baseline |
|---|---|---|---|---|---|---|---|
| t0860 | 197.68 ± 34.29 | **164.22 ± 10.28** | 206.20 ± 28.46 | 314.90 ± 29.66 | 230.45 ± 79.72 | 262.35 ± 66.88 | 414.41 ± 26.96 |
| t0868 | 131.42 ± 8.12 | **127.71 ± 14.26** | 178.45 ± 5.64 | 272.14 ± 26.34 | 191.08 ± 21.96 | 170.05 ± 27.26 | 411.98 ± 57.39 |
| t0869 | 106.69 ± 9.97 | 132.12 ± 31.37 | **104.47 ± 14.16** | 317.22 ± 12.75 | 244.38 ± 40.58 | 217.02 ± 57.01 | 393.12 ± 48.70 |
| t0872 | 144.11 ± 24.88 | 148.20 ± 23.63 | **134.48 ± 8.25** | 293.96 ± 19.00 | 221.13 ± 28.74 | 240.89 ± 24.17 | 374.48 ± 33.70 |
| t0879 | 89.00 ± 44.94 | 80.14 ± 16.21 | **64.63 ± 15.92** | 309.23 ± 69.40 | 172.41 ± 73.07 | 147.77 ± 15.72 | 364.79 ± 144.32 |
| t0900 | 193.74 ± 10.78 | 171.05 ± 25.41 | **158.56 ± 9.87** | 254.11 ± 18.63 | 209.07 ± 11.90 | 265.77 ± 79.99 | 399.16 ± 83.48 |
| t0912 | **113.00 ± 22.31** | 169.55 ± 27.35 | 150.70 ± 8.53 | 227.17 ± 22.11 | 192.28 ± 39.45 | 271.30 ± 28.89 | 406.25 ± 31.42 |
| t0920 | **80.46 ± 14.98** | 136.94 ± 36.43 | 84.83 ± 19.70 | 361.19 ± 71.25 | 261.72 ± 59.67 | 191.86 ± 37.85 | 398.22 ± 25.60 |
| t0921 | 187.89 ± 46.15 | 165.97 ± 42.39 | **142.97 ± 27.09** | 382.69 ± 20.27 | 260.49 ± 16.09 | 207.19 ± 24.84 | 363.92 ± 35.79 |
| t0922 | 254.83 ± 91.28 | **110.54 ± 43.99** | 227.73 ± 26.41 | 366.72 ± 8.10 | 290.71 ± 7.22 | 130.46 ± 11.64 | 419.14 ± 45.49 |
| t0942 | 188.55 ± 11.10 | 167.53 ± 22.01 | **137.21 ± 7.43** | 371.31 ± 9.90 | 233.78 ± 84.95 | 254.38 ± 47.21 | 393.03 ± 24.93 |
| t0944 | 146.59 ± 8.41 | **138.67 ± 50.36** | 245.79 ± 58.16 | 263.03 ± 9.43 | 199.40 ± 51.11 | 157.90 ± 2.57 | 404.12 ± 40.82 |

Table S5: Quantified distance between the empirically observed enzyme class exchange preferences of Cuesta et al. (2015) and the class exchange preferences inferred from LEGS-FIXED, LEGS-FCN, and a GCN. We measure the cosine distance between the graphs represented by the chord diagrams in Figure 2. As before, the self-affinities were discarded. LEGS-Fixed reproduces the exchange preferences the best, but LEGS-FCN still reproduces well and has significantly better classification accuracy.

| LEGS-FIXED | LEGS-FCN | GCN |
|---|---|---|
| 0.132 | 0.146 | 0.155 |

ensemble the learned features from a learnable scattering network (LEGS-FCN) with those of GCN and compare this to ensembling fixed scattering features with GCN as in Min et al. (2020), as well as the solo features. Our setting is slightly different in that we use the GCN features from pretrained networks, only training a small 2-layer ensembling network on the combined graph level features. This network consists of a batch norm layer, a 128 width fully connected layer, a leakyReLU activation, and a final classification layer down to the number of classes. In Table S6 we see that combining GCN features with fixed scattering features in LEGS-FIXED or learned scattering features in LEGS-FCN always helps classification. Learnable scattering features help more than fixed scattering features overall and particularly in the biochemical domain.

Table S6: Mean ± standard deviation test set accuracy on biochemical and social network datasets.

| | GCN | GCN-LEGS-FIXED | GCN-LEGS-FCN |
|---|---|---|---|
| DD | 67.82 ± 3.81 | **74.02 ± 2.79** | 73.34 ± 3.57 |
| ENZYMES | 31.33 ± 6.89 | 31.83 ± 6.78 | **35.83 ± 5.57** |
| MUTAG | 79.30 ± 9.66 | 82.46 ± 7.88 | **83.54 ± 9.39** |
| NCI1 | 60.80 ± 4.26 | 70.80 ± 2.27 | **72.21 ± 2.32** |
| NCI109 | 61.30 ± 2.99 | 68.82 ± 1.80 | **69.52 ± 1.99** |
| PROTEINS | 74.03 ± 3.20 | 73.94 ± 3.88 | **74.30 ± 3.41** |
| PTC | 56.34 ± 10.29 | **58.11 ± 6.06** | 56.64 ± 7.34 |
| COLLAB | 73.80 ± 1.73 | **76.60 ± 1.75** | 75.76 ± 1.83 |
| IMDB-BINARY | 47.40 ± 6.24 | 65.10 ± 3.75 | **65.90 ± 4.33** |
| IMDB-MULTI | 39.33 ± 3.13 | **39.93 ± 2.69** | 39.87 ± 2.24 |
| REDDIT-BINARY | 81.60 ± 2.32 | 86.90 ± 1.90 | **87.00 ± 2.36** |
| REDDIT-MULTI-12K | 42.57 ± 0.90 | 45.41 ± 1.24 | **45.55 ± 1.00** |
| REDDIT-MULTI-5K | 52.79 ± 2.11 | **53.87 ± 2.75** | 53.41 ± 3.07 |

Table S7: Mean ± std. over four runs of mean squared error over 19 targets for the QM9 dataset, lower is better.

| | LEGS-FCN | LEGS-FIXED | GCN | GraphSAGE | GIN | Baseline |
|---|---|---|---|---|---|---|
| Target 0 | **0.749 ± 0.025** | 0.761 ± 0.026 | 0.776 ± 0.021 | 0.876 ± 0.083 | 0.786 ± 0.032 | 0.985 ± 0.020 |
| Target 1 | **0.158 ± 0.014** | 0.164 ± 0.024 | 0.448 ± 0.007 | 0.555 ± 0.295 | 0.191 ± 0.060 | 0.593 ± 0.013 |
| Target 2 | **0.830 ± 0.016** | 0.856 ± 0.026 | 0.899 ± 0.051 | 0.961 ± 0.057 | 0.903 ± 0.033 | 0.982 ± 0.027 |
| Target 3 | 0.511 ± 0.012 | **0.508 ± 0.005** | 0.549 ± 0.010 | 0.688 ± 0.216 | 0.555 ± 0.006 | 0.805 ± 0.025 |
| Target 4 | **0.587 ± 0.007** | **0.587 ± 0.006** | 0.609 ± 0.009 | 0.755 ± 0.177 | 0.613 ± 0.013 | 0.792 ± 0.010 |
| Target 5 | **0.646 ± 0.013** | 0.674 ± 0.047 | 0.889 ± 0.014 | 0.882 ± 0.118 | 0.699 ± 0.033 | 0.833 ± 0.026 |
| Target 6 | 0.018 ± 0.012 | 0.020 ± 0.011 | 0.099 ± 0.011 | 0.321 ± 0.454 | **0.012 ± 0.006** | 0.468 ± 0.005 |
| Target 7 | 0.017 ± 0.005 | 0.024 ± 0.008 | 0.368 ± 0.015 | 0.532 ± 0.405 | **0.015 ± 0.005** | 0.379 ± 0.013 |
| Target 8 | 0.017 ± 0.005 | 0.024 ± 0.008 | 0.368 ± 0.015 | 0.532 ± 0.404 | **0.015 ± 0.005** | 0.378 ± 0.013 |
| Target 9 | 0.017 ± 0.005 | 0.024 ± 0.008 | 0.368 ± 0.015 | 0.532 ± 0.404 | **0.015 ± 0.005** | 0.378 ± 0.013 |
| Target 10 | 0.017 ± 0.005 | 0.024 ± 0.008 | 0.368 ± 0.015 | 0.533 ± 0.404 | **0.015 ± 0.005** | 0.380 ± 0.014 |
| Target 11 | **0.254 ± 0.013** | 0.279 ± 0.023 | 0.548 ± 0.023 | 0.617 ± 0.282 | 0.294 ± 0.003 | 0.631 ± 0.013 |
| Target 12 | 0.034 ± 0.014 | 0.033 ± 0.010 | 0.215 ± 0.009 | 0.356 ± 0.437 | **0.020 ± 0.002** | 0.478 ± 0.014 |
| Target 13 | 0.033 ± 0.014 | 0.033 ± 0.010 | 0.214 ± 0.009 | 0.356 ± 0.438 | **0.020 ± 0.002** | 0.478 ± 0.014 |
| Target 14 | 0.033 ± 0.014 | 0.033 ± 0.010 | 0.213 ± 0.009 | 0.355 ± 0.438 | **0.020 ± 0.002** | 0.478 ± 0.014 |
| Target 15 | 0.036 ± 0.014 | 0.036 ± 0.011 | 0.219 ± 0.009 | 0.359 ± 0.436 | **0.023 ± 0.002** | 0.479 ± 0.014 |
| Target 16 | 0.002 ± 0.002 | 0.001 ± 0.001 | 0.017 ± 0.034 | 0.012 ± 0.022 | **0.000 ± 0.000** | 0.033 ± 0.013 |
| Target 17 | 0.083 ± 0.047 | **0.079 ± 0.033** | 0.280 ± 0.354 | 0.264 ± 0.347 | 0.169 ± 0.206 | 0.205 ± 0.220 |
| Target 18 | **0.062 ± 0.005** | 0.176 ± 0.231 | 0.482 ± 0.753 | 0.470 ± 0.740 | 0.321 ± 0.507 | 0.368 ± 0.525 |

