# OpenReview forum: "Data-driven Learning of Geometric Scattering Networks"
_ICLR.cc/2021/Conference — Reject_

### Official Review · AnonReviewer4 · 2020-10-22
**Principled approch, minor contribution**

**Rating:** 4
**Confidence:** 3

**Review:**

Post discussion period:
-------------------------------
I read the other reviews and the author's response. Thank you for performing these extra experiments - they make the experimental section more complete. I still feel that the technical contribution of this paper is marginal and stick to my original rating.


-------------------------------
Summary:

The paper suggests a new graph neural network architecture based on a recently suggested wavelet-like transform for graph data called geometric scattering transform (Gao et al., 2019; Gama et al., 2019a; Zou & Lerman, 2019).  The Geometric Scattering Transform is a fixed feature extractor with a GNN structure and fixed wavelet filters that can extract both high and low-frequency features. The resulting features were shown to be effective in various graph analysis and learning tasks. Importantly, this transform relies heavily on two fixed choices: (1) A row -stochastic diffusion matrix P= ½ (I_n +WD^{-1}) that is used to extract features (2) A specific choice of dyadic frequency bands to partition the spectrum and governs the wavelet filter bank.

The main contribution of the paper is to relax these two hard-coded choices (1-2) and suggest a mechanism that will learn them from data. Concretely, for (1) the authors suggest adding a learnable parameter \alpha  that controls the laziness of the operator P, i.e. P=\alpha* I_n + (1-\alpha)WD^{-1}. For (2), the authors suggest learning (soft) frequency bands using a softmax mechanism. The authors then show that two useful properties of the original transform (stability, equivariance) are still valid The authors present results of their method on graph classification and regression tasks. The results indicate that the method performs well.



Strong points:

The idea of taking a fixed architecture that works well and relaxing it to have components that are learned from data is a good idea that has worked well before.
The approach is principled: (1) Take a good fixed extractor (2) Relax it to better represent the data (3) show that the good properties of the original transform still hold (4) compare to the original transform.

Weak points:

The main contribution of this paper, namely relaxing the laziness parameter and learning the frequency bands, is minor. I don’t think that it is sufficient for an ICLR paper.
The relaxation of the laziness parameter is not actually used throughout the paper since it did not work well.
The evaluation is rather limited, especially when considering the relatively minor novelty of the model. For example,  the authors do not compare to state of the art models such as GIN.

Recommendation:

 Although the idea of the paper makes sense, and the method seems to work, I believe its contribution is limited.


Minor comments:

- Equation 2 is difficult to parse, please explain all notation.
- Section 2 - it might be useful to exemplify/illustrate the transform on simple graphs.
- Lemma 1 - why self-adjointness is important?
- What is a nonexpansive frame?
- When learning the F matrix - how do you make sure there are no repeating rows?
-  “A somewhat less explored domain for GNNs is in biochemical graphs that represent molecules and tend to be overall smaller and less connected than social networks.” - I think that the datasets used in the paper are quite standard (DD, ENZYMES,...)

---

> ### Author Response · Authors · 2020-11-14
> **Thank you and response to AnonReviewer4**
>
> We thank the reviewer for their effort and thoughtful comments. In particular we appreciate the note on our principled approach, which we took care in implementing for a rigorous comparison on a wide range of datasets.
>
> Our main contribution is to explore a family of models based on graph scattering that inherit the theoretical properties of the fixed transform but are more flexible. We show that this model performs at least comparably to and in many cases better than standard models. Even if our model were to perform slightly worse than state of the art models, we believe it would be of interest given its theoretical soundness, and significantly fewer parameters.
>
> We recognize more comparisons would be helpful, and are happy to include comparisons to more methods such as GIN, Graph attention (GAT) and a more signal processing approach as mentioned by reviewer #1. We have preliminary results on GIN and GAT. Our results so far suggest that these two methods slightly outperform GCN, and GraphSAGE, but do not beat LEGS on the biological datasets. We expect to update the paper with full results by the end of the revision period.
>
> Thank you for your comments regarding equation 2 and more generally on the mathematical notation, we realize this nomenclature may not be used as widely outside the signal processing community. We are working on better explaining these terms before using them in section 3.
>
> Self-adjointness is important because it links models that use symmetric vs. asymmetric versions of the Laplacian or adjacency matrix. Lemma 1 shows that the diffusion matrix P (which is column normalized but not row normalized) is self-adjoint, as an operator, and can be considered a “symmetric” operator in a suitable inner product space. Since symmetric and asymmetric normalizations are used depending on the graph neural network literature, this link may be useful to the wider community as it links these normalizations theoretically.
>
> A non-expansive frame bound, as stated in the paragraph below Theorem 1 “implies stability in the sense that small perturbations in the input graph signal will only result in small perturbations in the representation extracted by the constructed filter bank.” This is in contrast to unconstrained networks where a small perturbation in the input space can have a large effect on the output, which is sometimes dealt with through weight regularization, clipping in Wasserstein GANs (Arjovsky et al. 2017), or gradient penalty (Gulrajani et al. 2017). Or in other words, inputs that are already similar in the considered L2 space will not explode to be much farther apart in the scattering features.
>
> We do not explicitly make sure there are no repeating rows, and experimentally this does not seem to be a large problem. A repeating row in F results in a feature that is (close) to all zeros, which is not very predictive, and so with an appropriate loss these similar rows will diverge.
>
> It is true that some of these biological datasets are becoming more standard in the literature, and on them we show we perform better than other methods (including GIN and GAT that will be added). However, many GNNs are still developed and tested on the semi-supervised node classification of social networks with the Cora, CiteSeer, and PubMed graphs. As Reviewer #2 mentioned, it is time to move beyond these benchmarks. GCN, GAT, GIN, Scattering-GIN (mentioned by reviewer #3), LancozNet and snowball (mentioned by Reviewer #1) are all developed with performance on these datasets in mind. Seeking performance on this benchmark biases the architecture design of most GNNs to work well on social networks. Architectures that are designed with biomedical graphs in mind are much less common.
> Beyond the more standard datasets, we also include a dataset that, to the best of our knowledge, does not appear in the machine learning literature - namely, the CASP dataset, where we show significant benefits over other GNNs. On ENZYMES, we go beyond the standard graph classification task, showing that our scattering representations better reflect the class exchange properties of ENZYMES with an orthogonal biological study of the underlying molecules. Thus, while we start with the more standard biological graph datasets for comparison, we also show marked improvement on less standard biological datasets and tasks.

---

> ### Author Response · Authors · 2020-11-25
> **Post revision update to AnonReviewer4**
>
> We have presented a theoretically sound graph neural network architecture that is based on the scattering transform rather than a GCN type structure. As mentioned in your review, this seems to perform surprisingly well given that it has many fewer parameters. We believe both the theory and empirical explorations performed in this work are of interest to the community.
>
> We have added comparisons to three state of the art models a graph attention network, a graph isomorphism network, and SnowballNet [Luan et al. 2020] that is similar to our model in that it takes a spectral perspective but is different in that it focuses on semi-supervised node classification. We find that the inclusion of these comparisons, and in particular GIN, which performs quite well, indeed help position our work and establish its contributions. We note that these additional results do not change the overall conclusions of our work, but rather strengthen them.
>
> We have added further explanations to the theoretical section particularly in terms of definitions and significance for audiences outside of the graph signal processing community.

---

### Official Review · AnonReviewer2 · 2020-10-26
**Notable contribution in learning the scales of graph scattering transforms.**

**Rating:** 8
**Confidence:** 5

**Review:**

Summary:

This paper proposes a novel graph neural network-based architecture. Building upon the theoretical success of graph scattering transforms, the authors propose to learn some aspects of it providing them with more flexibility to adapt to data (recall that graph scattering transforms are built on pre-designed graph wavelet filter banks and do not learn from data). By dropping the dyadic distribution of frequencies within the wavelet bank, the proposed architecture actually learns a more suitable frequency separation among the different wavelets.

Strong points:

The paper is well written and technically solid.

The idea of flexibilizing scattering transforms while retaining key theoretical contributions is a notable contribution. In particular, the idea of learning the dyadic nature of the frequency bands in the wavelet bank is a key aspect, since the frequencies in graphs are rarely evenly spaced, and thus the ability to partition the space of frequencies in other form than a dyadic one is bound to bring substantial improvements.

The numerical experiments in the paper deal with new biochemical datasets which are much more interesting and practically useful than the de-facto benchmarks of semi-supervised learning in citation networks. This is a welcome change to see actually useful applications of graph neural networks.

Weak points:

No real weak points except for a common misunderstanding of the low-pass nature of filters in a graph convolutional neural network. Please, see below for details. I seriously encourage the authors to re-write parts of the abstract and introduction to avoid perpetuating the misconception that graph convolutional neural networks can only learn low-pass filters.

Recommendation:

This is a very interesting contribution and, provided the clarifications that I mention below, I firmly support the acceptance of this paper into ICLR.

Major comment:

The claim that GCNs rely on low-pass graph filters is, at the very least, misleading. As a matter of fact, even a graph convolution with an order-one polynomial can be a high-pass. To see this, assume we adopt the graph Laplacian as the matrix description of the graph. If this is the case, we know that low-eigenvalues correspond to low-frequencies and high-eigenvalues to high-frequencies (according to the notion of total variation). If this is the case, then a filter like H(L) = 0*I+1*L gives a frequency response of h(lambda) = lambda which is 0 for the zero eigenvalue, and grows for larger eigenvalues. This is an example of a high-pass filter. Likewise, assume that we use the adjacency matrix. If this is the case, then the highest real eigenvalue is the one with the lowest frequency, and depending on how far we are from that eigenvalue (measured as modulus operation in the complex plane), the higher the frequency is. So, for the sake of argument, let us assume that the adjacency matrix has real eigenvalues and is normalized by the largest eigenvalue, so that all eigenvalues are contained in [-1,1]. For this situation, eigenvalues closest to 1 will be low frequencies, and eigenvalues closest to -1 will be high frequencies. Then, a filter of the form H(A) = 0.5*I - 0.5*A gives a frequency response of h(lambda) = 0.5 * (1-lambda) which has value 0 for the lowest frequency, and value 1 for the highest possible frequency. This makes such a filter a high-pass filter. As we can see, we have just constructed two filters (one for the Laplacian and one for the adjacency) where both filters are of order one, and still high-pass filters. Of course, it becomes easier to build high-pass filters if the orders of the polynomials are higher (i.e. ChebNets). Therefore, since the coefficients of the graph convolutions are learned from data, it is very hard to argue that the resulting filters will be low pass (i.e. they need not be, there is nothing preventing an order one graph filter to be a high-pass and there is no enforcement of such a constraint in the original formulation of graph convolutional neural networks).

This claim is first found in the abstract, and later repeated in the introduction. With the clarifications from the introduction, I may understand where the misinterpretation arises. Kipf's GCNs use as graph matrix a normalization of (I+A) and consider an order one polynomial with the zeroth coefficient set to zero: h*(I+A). This forces h_0 = h_1 in an order-one graph filter on the adjacency matrix, which would otherwise be written as h_0 * I + h_1 * A. By forcing h_0 = h_1, we are indeed forcing the filters to be low-pass, i.e., we always learn filters of the form h(lambda) = h*(1+lambda) where we only learn the coefficient h. However, this is a design choice of Kipf's GCNs. The general formulation of a graph convolutional neural network (see Defferrard's ChebNets, for instance), does not require h_0 = h_1 and thus can also learn high-pass filters as discussed above. Interestingly enough, even if we would consider the normalization of (I+A) as the graph matrix, but we don't force the zeroth order coefficient to be zero, we would be able to have filters of the form h_0 * I + h_1 * (I+A) which would actually allow the learned filters to be high pass (i.e. h_0 = 1, h_1 = -0.5, see above). In any case, what I mean here, is that learning low-pass filters is a self-inflicted problem by Kipf's GCNs that can be very easily avoided by just using a more general formulation of graph convolutional neural networks (for instance, Defferrard's ChebNets).

In summary, I would suggest the authors to avoid any mention that graph convolutional neural networks only learn low-pass filters. Otherwise, this would perpetuate an important misunderstanding that has been around for a while now and is misleading research efforts. In other words, yes, Kipf's 'GCN' implementation of graph convolutional neural networks can only learn low-pass filters. However, the general definition of graph convolutional neural networks (suggested in Defferrard's ChebNets, and formalized in the signal processing literature regarding GNNs) by no means implies that the learned filters are low-pass. Thus, 'graph convolutional networks' are not oversmoothers, or only learn low-pass filters. Kipf's GCN does (and it can be avoided by just using a different implementation).

In any case, this does not alter the contribution of the paper. I would just request the authors to correct this aspect to avoid perpetuating the misunderstanding surrounding graph convolutional neural networks.

Minor comments:

1) The authors suggest two learnable adaptations of graph scattering transforms. The parameter alpha in the graph matrix, and the scales. However, the authors find out that the parameter alpha does not contribute anything in training and is thus arbitrarily set to 1/2. I would suggest, then, that the authors focus on the scales as the learnable adaptation, and only mention in passing that alpha can be potentially trained, but that the numerical experiments in this paper show no improvement by doing so. I believe this would put the focus and draw attention to, probably, the main contribution of the paper.

2) When referring to GCNs, the authors cite Kipf's paper, Veličković's paper and Abu-El-Haija. The GAT architecture in Veličkovic's paper is not a graph convolutional neural network, so it shouldn't be cited here (GATs are, probably, the first case of a popular graph neural network architecture that is not convolutional). Also, I believe it is unfair not to cite the two main contributions to GCNs: Bruna's 'Deep Spectral Networks' paper from ICLR 2014, and Defferrard's 'ChebNets' from NeurIPS 2016. (Omission of ChebNets would explain the misunderstanding with respect to the graph convolutional neural networks being low-pass filters).

---

> ### Author Response · Authors · 2020-11-14
> **Thank you and response to AnonReviewer2**
>
> We thank you for your comprehensive and well reasoned review! We agree that whole graph classification is a much more interesting problem than the semi-supervised citation networks more commonly used.
>
> We agree that the way low-pass is used in this paper can be confusing and we will remove this terminology from the paper. We believe that this stems from the usage of GCN to mean the popular “Kipf’s GCN” in some circles. It certainly would make more sense if GCN referred to the original and more general formulation, and as you note these more general GCNs have no problem learning other types of filters.
>
> We agree the main contribution of this paper is the learning of scales. We will restructure the paper to focus more on the learning of scales, and present the \alpha and the graph specific filter selection (mentioned by reviewer 1), which both fit into the theoretical framework but do not show much numerical improvement, as theoretically possible in section 3, but not numerically beneficial at this time.
>
> We will correct these citation issues.

---

> ### Author Response · Authors · 2020-11-25
> **Post revision update to AnonReviewer2**
>
> Thank you again for the positive comments and for your clarification on the low-pass terminology. To address this specific concern, we have removed the low-pass terminology from the paper and instead explicitly separate the specific GCN architecture of Kipf and Welling from more general formulations of graph neural networks and graph convolutional networks.
>
> We have also put more emphasis on the main contribution -- that we are learning scattering scales -- as suggested.

---

### Official Review · AnonReviewer3 · 2020-10-27
**Official Blind Review #3**

**Rating:** 6
**Confidence:** 3

**Review:**

This paper extends geometric scattering network by relaxing its scattering construction to enable training / data-driven learning. There are three major modules in the proposed network architecture: diffusion module, scatter module, and aggregation module. They conduct experiments on two tasks: whole graph classification and graph regression.

The idea to relax geometric scattering network is novel to the best of my knowledge. However, I have the following concerns:

* Why is it that LEGS only outperforms other GCN methods on biological datasets? The major advantage of LEGS compared with other low-pass filter based GCNs is that it goes beyond low frequencies and consider richer notions of regularity. Why doesn't this advantage manifest on performances on other types of graph data (e.g. social networks)?

* The result in Table 2 does not seem promising. If LEGS only performs well on graphs that exhibit certain properties, showing results on synthetic datasets would help.

* I suggest that authors should report results on larger datasets like QM9. All experiments are conducted on datasets with no more than 5000 instances. Or is that due to computational complexity and scalability issues?

* What are the advantages of the proposed method when compared with scattering-GCN [1]?
They also address the problem of oversmoothing and scattering-GCN is also learned in a data-driven fashion. How does scattering-GCN  perform if we obtain whole-graph features by aggregating node features obtained by their method? Why is it not included in the baseline?

[1] Yimeng Min, Frederik Wenkel, and Guy Wolf. Scattering gcn: Overcoming oversmoothness in graph convolutional networks. arXiv preprint arXiv:2003.08414, 2020.

---

> ### Author Response · Authors · 2020-11-14
> **Thank you and response to AnonReviewer3 - Part 1/2**
>
> We thank the reviewer for their effort and thoughtful comments. As the reviewer notes, fixed geometric scattering features are used in Feng et al. 2019 and Min et al. 2020 to improve features found in standard GCN architectures. We also believe relaxing these fixed features is novel and can lead to improved performance with fewer parameters especially in biological domains. Now to address your specific comments:
>
> (1) We respectfully disagree with the assertion that Table 2 does not look promising. On 4/7 biological datasets a version of our model achieves the highest performance, and on the other three datasets our model is close behind, either second best or within one standard deviation of the best performing model.  On social graph datasets our model has the best or second best performance on all datasets excluding GS-SVM, which uses handcrafted non-learned features, and in particular scales. It seems these were tuned in [Gao et al. 2019] with emphasis on the social network datasets, which formed the main results table in that work (while biochemistry datasets were only presented in the supplement). Further, we note that other GNNs have highly varied performances. For example, GraphSAGE has the best performance on Collab, but does much worse on IMDB-Binary and Reddit-Binary. GCN, on the other hand, does better on Reddit-Multi, but worse on Collab and IMDB-Binary. Our model is consistent across datasets, and we will add a metric of average performance over datasets to emphasize this point. While for any specific dataset network various models can be tuned to improve performance (although some more than others, as shown in our table), our model is designed with relatively few parameters, thus simplifying this tuning task, and performs consistently across all datasets, thus alleviating model selection to some extent. This consistency is especially important when trying to derive better understanding from the structure of the dataset. We demonstrate this on Enzymes and on CASP, where we show we can do more intricate tasks than classification to better understand the structure and interaction between classes, which is crucial in biomedical applications, and indeed verify our results with respect to known biochemical properties - not only ones provided with the data itself, but also ones found in application-domain literature (e.g., EC exchange preferences in enzyme evolution).
>
> (2) We agree that comparisons on a larger dataset would be interesting. We note that the scalability of such comparisons does not only depend on the computational load of our model, but also that of other models we compare against. We will try to do such a comparison before the end of the discussion period, but even if this comparison is not complete, we do intend to incorporate it in the final version. We expect these additional results on currently available datasets (e.g. QM9 mentioned by the reviewer) to follow the trend discussed in the paper, and from our preliminary results do not anticipate that they will significantly affect our conclusions on where the LEGS model is most effective.

---

> > ### Author Response · Authors · 2020-11-14
> > **Thank you and response to AnonReviewer3 - Part 2/2**
> >
> > (3) Thank you for bringing up this interesting work! Scattering-GCN focuses on node-level classification, and does not consider graph-level features. There are many node representation learning methods and each of them can be aggregated in various ways. In contrast, our method focuses on graph-level features, and learning the scales of scattering. Further, Scattering-GCN computes node level features from GCN and from a fixed scattering transform, essentially ensembling these two models for better semi-supervised node classification than either model individually. The scattering channels there use fixed scales that are configured manually, or with a grid search that the authors detail in their supplement. Here, as a major component of this work, we learn the scattering scales from the data as part of the training rather than as hyperparameters. In general, ensembling methods such as Scattering-GCN, when done carefully, can often improve accuracy (at least slightly), but are orthogonal to our work. We improve the scattering features by learning them, and our scattering features could be combined with GCN (or features from any other model for that matter) before or after node aggregation. Such ensembling is perhaps likely to improve performance on the graph level tasks, just like it appears to do in node level tasks, but we believe this would be out of scope here and leave such investigation to future work.
> >
> > Based on these points, and since the ideas of scattering-GCN in generalized form can anyway probably be used in future work to improve the performance of graph level scattering, including LEGS, by ensembling it with GCN (or other) models, it is not a directly useful baseline, considering we already compare to both GCN and geometric scattering separately. However, for completeness, we will discuss this approach and better refer to node level networks and graph level features. As mentioned above, we leave the study of hybrid models (whether based on scattering-GCN or others) to future work.

---

> ### Author Response · Authors · 2020-11-25
> **Post revision update to AnonReviewer3**
>
> Thank you for the comments. We have added experiments on the QM9 dataset. We find that in line with experiments on the CASP graph regression task, LEGSNet often shows the greatest improvement on difficult-to-predict portions of the dataset. Whereas networks like GIN are able to perform slightly better on easier regression tasks, the learnable scattering features provide the flexibility and robustness to better approximate more difficult ones.
>
> We have added additional sections on the social graph datasets. Except for the fixed scattering transform of [Gao et al. 2020], which has been constructed with these social datasets in mind, LEGS-Net performs comparably if not slightly better than standard GNN models. Performing the best on three out of six datasets and second best on the other three, whereas the other GNN models are much less consistent.
>
> We also added an exploration of ensembling based approaches similar to Scattering-GCN. Scattering-GCN explored a combination model of fixed scattering with GCN. We compare this to a combination of features from learned scattering and GCN and find that on most datasets learned scattering helps more than fixed scattering, and we find that similar to the main graph classification task, this is especially apparent on biochemical datasets.

---

### Official Review · AnonReviewer1 · 2020-10-28
**Interesting connections between GSP and GNNs but with a few issues**

**Rating:** 6
**Confidence:** 4

**Review:**

Summary:

This paper proposes a parameterization of the scatter transform on graphs and builds graph neural networks based on this parameterization. Authors also demonstrate that this scatter transform could theoretically lead to stable hidden representations of GNNs. Experimental results on biochemical datasets are conducted to support the arguments of the paper.

Pros:

1, The connection between the recent advance of graph signal processing (i.e., the generalization of the scatter transform on graphs) and graph neural networks are inspiring and interesting.

2, I agree with the motivation of designing band-pass or high-pass graph convolutional filters since the original ones used by GCNs indeed act like low-pass filters and tend to over-smooth signals on graphs. Although I am not so sure whether the scatter transform is the way to go to design data-driven/learnable graph convolutional filters which satisfy this purpose, the exploration along this line is still valuable and could inspire others in the community.

3, Theorem 1 is interesting since it reveals the stability of hidden representations of the hand-crafted GNN.

4, The overall paper is written in a clear manner.

5, Authors provide experimental results on a wide range of datasets.

Cons & Questions & Suggestions:

1, If I understood correctly, Theorem 1 holds for the construction shown in Eq. (1). However, it is unclear whether Theorem 1 still holds for the relaxed construction of filter banks. This part is not discussed in the paper. If the theoretical guarantees on the stability do not hold anymore, then the contribution of the proposed method will be degraded significantly.

2, I’m concerned about the emphasis on significantly fewer learned parameters of the proposed model. If the performances of your proposed model are on par with or superior to other GNNs on many datasets, then having fewer parameters would be a merit. However, on common social networks, as shown in the appendix, the performance of the proposed model is worse than other basic GNNs, not mentioning the recent ones. This also links to the somewhat less convincing experimental comparison on biomedical datasets as discussed in point 6. On the other hand, it seems that there are a few options to increase the number of parameters (as discussed in point 4) which arguably could improve the model capacity.

3, Some important references on the intersection of the graph signal processing and graph neural networks are missing, e.g., [1,2,3]. In particular, it would be great to discuss the relationship between your work and [2,3] which also go beyond low-pass filter on graphs by exploiting learnable spectral filters and capturing the multi-scale diffusion. Comparing the scatter transform with their approaches would help better position your contribution.

4, Regarding the design choice of learning the selection index F, it is less expressive to share the selection index among different graphs. In other words, the current design of using the same set of \theta for all graphs is less satisfying as one can imagine that some graphs may require capturing long-range dependencies (corresponding to the higher power of diffusion matrix P) whereas some may just require short-range dependencies. A better option would be designing a model that takes graph data as input and predicts the selection index. Therefore, the selection index would be graph-dependent. Moreover, compared to [2,3], if one wants to explore larger diffusion steps, the computational cost would be much higher than [2,3] since they avoid the direct computation of matrix powers by using the approximated top eigenpairs of the diffusion matrix P.

5, For the aggregation part, there are quite a few simple options like max/mean/attention. It would be more convincing to include these options and conduct an ablation study to justify the design choices like the RBF aggregation.

6, In the experiment section, it would be more convincing to add comparisons with some recent competitive methods, especially those motivated from the graph signal processing like [2,3]. Could you shed some light on why GraphSAGE has similar std as your method and other baselines on graph classification tasks while significantly larger std than yours and others on graph regression tasks? Also, why does the proposed LEGS-RBF have a huge performance drop on the MUTAG dataset compared to any other baselines? It seems that the proposed methods perform comparable or better than other baselines on biochemical datasets but worse on social science datasets (as shown in the appendix). Could you explain why this is the case?

[1] Defferrard, M., Bresson, X. and Vandergheynst, P., 2016. Convolutional neural networks on graphs with fast localized spectral filtering. In NeurIPS.

[2] Liao, R., Zhao, Z., Urtasun, R. and Zemel, R.S., 2019. Lanczosnet: Multi-scale deep graph convolutional networks. In ICLR.

[3] Luan, S., Zhao, M., Chang, X.W. and Precup, D., 2019. Break the ceiling: Stronger multi-scale deep graph convolutional networks. In NeurIPS.

Conclusion: Based on the above merits and issues, I am currently on the borderline and would like to hear feedback from the authors.

===================================================================================================

The response and the updated version clarify and address many of my concerns regarding contributions and empirical conclusions. Overall, I lean towards acceptance.

---

> ### Author Response · Authors · 2020-11-14
> **Thank you and response to AnonReviewer1 - Part 1/2**
>
> We thank the reviewer for their thoughtful comments and suggestions. Thank you for noting our wide range of experiments, we believe it is important to show all results including on datasets where our method did not perform the best. Now to address your specific concerns:
> (1)  Theorem 1 holds for the relaxed construction. We note that in the paragraph preceding theorem 1 we mention that “The following theorem shows that for any selection of scales, the relaxed construction of WJ′ constructs a nonexpansive frame, similar to the result from Perlmutter et al. (2019) shown for the original handcrafted construction.” We will clear up the wording here.
>
> (2)  Table S2 shows that our model outperforms other GNNs on some biomedical benchmarks and that it performs comparably on social network datasets. Out of the six social network datasets, ignoring the fixed scattering model GS-SVM, our model outperforms both GNN models on 3 of them, and is second best on the other three. This is at least comparable if not slightly superior. We provide a model with significantly fewer parameters that achieves comparable or superior accuracy on common benchmarks. Even when our method shows comparable results, and definitely when it outperforms other GNNs, we believe that its smaller number of parameters could be useful in applications with limited compute or limited training examples. Based on comments from the reviewer it seems this point was not clear enough in the paper and we will work to clarify it. One source of this confusion may have been in our paragraph headed by “LEGS outperforms on biological datasets”, where we said “In particular we find that LEGSNet outperforms other methods by a significant margin on biochemical datasets with relatively small but high diameter graphs (NCI1, NCI109, ENZYMES, PTC), as shown in Table 2, whereas on the social network datasets GCN and GraphSage perform quite well”. We were trying to emphasize that we believe our model will be particularly beneficial on graphs that are similar to biochemical type graphs. This is not not to say that our model performs poorly on social network datasets, but merely that other methods have comparable results there, and our main motivation was not driven necessarily by such applications but rather by the biochemistry ones. We will change the wording in this paragraph to make this clearer.
>
> (3)  Thank you for bringing these works to our attention, we will include them in related work. [2] provides an interesting way to approximate the spectrum of the graph which is useful for large scales and could be especially useful for large graphs, where polynomial filters on the Laplacian are cumbersome. [3] tackles the oversmoothing problem present in Kipf and Welling GCNs on semi-supervised node classification (rather than whole-graph representation considered here). Their “snowball” is similar to the GraphSAGE network that we use, where both effectively have skip connections that enable concatenating features from all levels before classification. We believe these multiscale features are key. However, concatenating them (as in GraphSAGE, which we compare to here) provides many redundant features especially at longer or wider scales; we tackle these redundancies (at least to some extent) by learning the scales instead of concatenating all of them.
>
> (4)  Learning the selection indices on a per-graph basis is an excellent idea! This could be thought of as some sort of self-attention on each graph to learn F. We have tried a few attention based models where F depends on the graph, but were unable to find conclusive results. We believe this may have to do with the finicky nature of attention, which is not trivial to train, and this type of model may show improvement on larger datasets or with a more extensive parameter search. This is certainly an interesting direction to keep pursuing, but at this point we believe it is somewhat out of scope for this paper, and we leave this to future work. We thank the reviewers again for their excellent suggestion.

---

> > ### Author Response · Authors · 2020-11-14
> > **Thank you and response to AnonReviewer1 - Part 2/2**
> >
> > (5)  We note that the RBF network we used operates on graph-level permutation invariant (i.e., already aggregated, node-order independent) features, so this is, in fact, not an aggregation method. The aggregation function used here to transform permutation equivariant node-level features to permutation invariant graph-level ones is interesting, but it is not the main focus of our work. We use here the same aggregation method (i.e., using four statistical moments - mean, variance, skewness, kurtosis) presented in geometric scattering (Gao et al., 2019), which seems particularly effective there, and is also effective in our case. That being said, for the purpose of relaxing the construction of geometric scattering, we agree it could be interesting to also consider alternative aggregation methods via an ablation study, which we will add (at least in the appendix), comparing the statistical-moment aggregation with other commonly used methods (max, top-k, etc. - we note that “mean” is already included as the first statistical moment) to shed light on whether a different aggregation could be more effective in our slightly relaxed setting.
> >
> > (6)  To answer each of your separate questions here:
> > *  Comparisons: We plan to add comparisons to more recent methods such as the two you mention, as well as GIN and GAT. We already have partial results on GIN and GAT, as expected these methods slightly outperform simpler GNNs such as GCN and GraphSAGE, but do not outperform LEGS, and do not expect that these will alter our conclusions.
> > *  The difference in standard deviation of GraphSAGE: This is an intriguing observation. While it is impossible to know for sure, we believe could be because GraphSAGE by default uses a max aggregation, which may be more unstable for a regression type task than a classification task. However, we note that thorough study of the inner working of GraphSAGE is quite far from the scope of this work, and therefore providing clear insights about how well (and why) that method would perform on various tasks is probably best left for future investigation.
> > *  Performance of LEGS-RBF on the MUTAG dataset: MUTAG is extremely small, consisting of 188 graphs, which means that our validation / test set sizes are ~18 graphs each. The RBF network can overfit here and the test set may be far from any of the learned centers, leading to poor performance. We note that even in limited data settings, 188 graphs is very small for neural networks, and realistically we expect most datasets encountered to be an order of magnitude bigger than MUTAG.
> > *  Performance on biomedical vs. social graphs: As mentioned above in paragraph (2) we respectfully disagree with the assertion that LEGS doesn’t perform well on social graph datasets. It is true that, as we show, LEGS performs particularly well on small but high diameter graphs like those found in the biomedical domain, but it also performs comparably on social network graphs as a whole (even though particular networks may be better in particular cases). Looking at table S2, out of the six social network datasets, ignoring the fixed scattering model GS-SVM, our model outperforms both GNN models on 3 of them, and is second best on the other three. GraphSAGE does a bit better on Collab, but much worse on IMDB-Binary and Reddit-Binary. GCN does a bit better on Reddit-Multi, but worse on Collab, IMDB-Binary, and Reddit-Binary. From our preliminary experiments, GIN does slightly better than these GNNs, for example, on IMDB-Binary and (in biochemistry data) on Enzymes, but not better than LEGS. Overall, LEGS is more stable than these other GNNs - ranking first or second best over most of the datasets. We will add a measure of the average performance across datasets to emphasize this point. We note that we mainly focus in this work on the biomedical domain where the benefits of LEGS are clearer, but provide the social graph comparison for completeness. We find the observed benefits from LEGS are exhibited more clearly on graphs with large diameter relative to their size as found in biomedical graphs. While we do not focus on social graphs (again, these are shown mainly for completeness), we note that these graphs are substantially different as they are primarily ego graphs, with relatively small diameters compared to their sizes.

---

> ### Author Response · Authors · 2020-11-25
> **Post revision update to AnonReviewer1**
>
> Thank you for your time and helpful comments in improving this work. In the revised manuscript we have provided three additional comparisons to GIN, GAT, and SnowballNet [Luan et al. 2020]. We chose SnowballNet over LancozNet to compare to as it outperformed LancozNet and AdaLancozNet in [Luan et al. 2020]. We found that SnowballNet was highly optimized for the semi-supervised node classification task and did not perform very well in the whole graph classification task. Since it is optimized for a single graph, running their code expired our time limit of 10 hours per model on a number of datasets. We will explore longer time limits in the final version but do not anticipate this will significantly affect results.
>
> We added a section in the results and in the appendix on the performance of LEGSNet on social network datasets. In summary, our model performs well except that the fixed scattering transform of Gao et al 2020 performs slightly better, which is expected as it has been constructed with these datasets in mind.
>
> We are still in the process of performing an ablation study exploring pooling strategies and we thank you for the suggestion. This will either be added in the appendix of the final version or be studied more extensively in future work.

---

### Author Response · Authors · 2020-11-25
**Thank you and summary of revisions**

We thank the reviewers for their positive feedback especially in regard to the importance of learning the scattering scales. We also appreciate their specific comments which we have used to improve this paper.
In summary, previous work on geometric scattering [Zou and Lerman 2019, Gama et al. 2019, Gao et al. 2019] established the viability of fixed geometric scattering for whole graph classification. In this work we expand on this exploring the space between fixed scattering transforms and fully flexible GNNs. We learn the scales of geometric scattering such that the model is still theoretically sound, while also demonstrating empirical improvements over fixed geometric scattering and recent GNN models on graph classification and regression, focusing in the biochemical domain, where graphs are notably smaller and higher diameter than those in the social network domain.
We have focused on the following items for the revision as well as improving the writing throughout:
1. Added comparisons to three more state of the art models (GIN, GAT, and SnowballNet). We thank the reviewers for their suggestions and believe these comparisons help place LEGSNet relative to more recent and powerful models. While these models perform well (in particular GIN on our data) they do not change the conclusions of this work. Namely that with significantly fewer parameters, learnable geometric scattering performs well on the graph classification task specifically on biomedical graphs.
2. Added evaluation on the larger QM9 dataset. We find that LEGS-FCN performs the best which is consistent with the CASP regression task.
3. Evaluated the effect of ensembling scattering with GCN as Min et al. 2020 showed on node classification. We find that consistent with Min et. al 2020, ensembling also helps in graph classification, and that ensembling learnable scattering features with GCN is more beneficial than fixed scattering features with GCN.
4. Added a section in the results regarding performance on social network datasets. LEGSNet performs the most consistently well excluding GS-SVM where the fixed features were designed with these specific datasets in mind.
5. Cleared up the behavior of general GCNs vs. Kipf and Welling’s GCN with regard to low-pass filters.
6. Added relevant citations as pointed out by the reviewers.

Given feedback from the reviewers we believe this work is interesting to the community and that our additional experiments in this revision should satisfy the reviewers’ empirical concerns.

---

### Decision · Program_Chairs · 2021-01-07
**Final Decision**

**Decision:**

Reject

**Comment:**

This paper proposes a simple yet powerful generalisation of graph scattering transforms that allows a flexible scale dilation structure, retaining the stability guarantees of dyadic transforms. Experiments with strong empirical performance are reported on a variety of biochemical tasks.

Reviewers acknowledged the soundness of the approach as well as the quality of the empirical evaluation, but also raised some concerns about lack of novelty. Ultimately this AC believes that, although this work solidifies Graph Scattering Transforms as a good alternative to GNNs on certain structured physical domains, it provides little advancements on the theory front. Unfortunately not all good papers can be accepted, and therefore the AC recommends rejection at this time, encouraging a resubmission.